# Inter-laboratory ring trial to compare four quantitative polymerase chain reaction assays employed for detection of *Mycobacterium avium* subspecies *paratuberculosis*

L. Worsley,[1] P. L. Davies[1]

**ABSTRACT**  Johne's disease is an infectious enteric disease caused by *Mycobacterium avium* subspecies *paratuberculosi*s (MAP) affecting ruminant species worldwide. In Project 1, an independent performance comparison ring trail was conducted between three different commercial MAP quantitative polymerase chain reaction (qPCR) assay services (B, C, and D) currently marketed in Great Britain by three separate laboratories against each other and against a fourth assay (A) not available commercially in Great Britain. A total of 205 individual ovine and bovine samples from five farms were analyzed to give 41 sets of pooled results (pool size five) from each laboratory according to their specific protocols. The numbers of positive pools for assays A–D were 18, 12, 11, and 1 (43.9%, 29.2%, 26.8%, and 2.4%), respectively. Assessment of interrater reliability produced a Fleiss' kappa coefficient of 0.15, indicating very poor overall agreement between the four laboratories. Laboratories A–D diagnosed 4, 3, 2, and 1 flocks at the farm level, respectively, as MAP positive. In Project 2, 38 pooled ovine samples from 10 flocks were analyzed to compare the performance of laboratories A and B. The numbers of positive results for laboratories A and B were 24 (63.1%) and 17 (44.7%), respectively (Cohen's kappa 0.54), indicating that laboratory A was more sensitive than B in line with results from Project 1. Variation between laboratories offering MAP qPCR assays is a significant concern, and further work is warranted to validate and standardize the performance of assays between laboratories for both ovine and bovine samples.

**IMPORTANCE**  Our study reports the findings of an inter-laboratory ring trial comparing the performance of four different quantitative polymerase chain reaction (qPCR) assay services for detecting *Mycobacterium avium* subspecies *paratuberculosis* (MAP) infection in cattle and sheep. MAP is the causative agent of Johne's disease (also known as paratuberculosis), a significant production-limiting disease in livestock populations with a worldwide distribution. The content of this paper is significant and novel as it is the first to highlight the marked variation between the diagnostic sensitivity and reproducibility of the three principal commercial laboratories offering MAP qPCR diagnostic and screening services in Great Britain. The low sensitivity and high variability between the laboratories are of great concern and relevance to veterinary practitioners and livestock producers.

**KEYWORDS**    ring trial, paratuberculosis, Johne's, ovine, bovine

Johne's disease (JD), or paratuberculosis, is an important production-limiting chronic wasting disease caused by *Mycobacterium avium* subspecies *paratuberculosis* (MAP), affecting predominantly ruminants with a worldwide distribution. The disease is chronic and insidious, with a long incubation period during which infected animals shed MAP in their feces intermittently and at low levels initially (1, 2), progressing to persistent and

**Ad Hoc Peer Reviewers** Fazeela Arshad, National Institute for Biotechnology and Genetic Engineering, Faisalabad, Punjab, Pakistan; Asra'a Abdul-Jalil, University of Anbar, Sulaimaniya, Iraq; Muthuraj Muthaiah, Government Hospital for Chest Diseases, Puducherry, Puducherry, India

Address correspondence to L. Worsley, L.Taylor11@liverpool.ac.uk.

The authors declare no conflict of interest.

See the funding table on p. 12.

heavy shedding as the disease progresses towards the clinical stages for those animals that are not capable of clearing the infection (3, 4). In addition to negatively impacting animal welfare, JD is responsible for economic losses through reduced productivity, and increased replacement costs associated with premature culling and increased mortality (5, 6). To this end, disease control programs have been developed in many countries, including Great Britain (7). These programs require robust and reliable diagnostic and screening tests to establish herd or flock MAP infection status, to identify infected individual animals for culling, and make informed decisions regarding retention of their progeny, identify uninfected herds and flocks that are free from disease as sources of replacement stock, and also to satisfy specific health scheme requirements. Paratuberculosis also raises public health concerns with ongoing contention around MAP's suspected role in Crohn's disease in humans. This emphasizes the importance of implementing on-farm JD control programs (8).

Antemortem diagnosis of MAP infection in the early sub-clinical stages of the disease is challenging due to the lack of sufficiently sensitive and cost-effective tests. Additionally, while commercially available MAP quantitative polymerase chain reaction (qPCR) assays are validated for bovine samples in GB, the same laboratories do not offer validated equivalent tests for ovine samples. The consensus is that fecal qPCR and culture are more sensitive than antibody tests, especially in the early stages of the disease as MAP fecal shedding precedes a detectable antibody response in sheep and cattle (9–11).

Antibody enzyme-linked immunosorbent assays (ELISA) are most appropriate in the more advanced stages of disease associated with increasingly discernible clinical signs (1). However, despite their low sensitivity in the early stages of infection, ELISA assays are frequently employed as herd and flock screening tests because of their practicality and low cost. Indeed, whole herd serological screening using ELISA tests is routinely used for official JD control programs for GB beef herds and, less commonly, sheep flocks.

While fecal culture is still often considered the "gold standard" mainly due to its very high specificity (12), sensitivity estimates vary considerably depending on the MAP strain in question and stage of infection and, thus, the magnitude of fecal shedding in the individual or population being tested (13, 14). In contrast to C-strains which are relatively amenable to laboratory culture techniques, S-strains display slow and fastidious growth and are highly susceptible to various decontamination procedures and certain antibiotics (15, 16). As a result, it's plausible that the sensitivity of fecal culture is underestimated where S strains predominate (14). How relevant this is to GB is unknown, given the lack of information regarding circulating MAP strains in our sheep and cattle populations. Fecal real-time qPCR assays have been shown to demonstrate superior sensitivity to fecal culture (14, 17–19). Plausible explanations for this include the ability of qPCR to detect both C- and S- strains and also live and dead MAP organisms (14).

To the author's knowledge, there have been no published independent trials comparing the performance of the different MAP qPCR assays commercially available in GB. This study aimed to establish the comparable performance of the pooled fecal MAP qPCR services that are commercially available as screening tests for veterinarians in clinical practice. The inter-laboratory ring trial study design encompasses the combined effect of intrinsic biochemical and extrinsic procedural factors that influence the sensitivity and specificity of the testing service provided to the clinician.

## MATERIALS AND METHODS

Two projects were conducted as part of this inter-laboratory ring trial study. Project 1 compared the performance of three pooled fecal MAP qPCR tests (assay services B, C, and D) as provided by the three prominent laboratories (B, C, and D, respectively) offering this assay in GB against each other and also against a fourth qPCR (A) (20) not currently available commercially in GB and performed at the University of Liverpool (laboratory A). Laboratories B and C are government subsidized, while laboratory D is a private commercial enterprise. Project 2 compared the performance of laboratory A against laboratory B using an additional set of samples.

**TABLE 1** Summary of total number of ovine and bovine pools analyzed from farms V–Z

| Farm ID | Number of ovine pools tested | Number of bovine pools tested | Total number of pools tested |
|---|---|---|---|
| V | 5 | 2 | 7 |
| W | 5 | 0 | 5 |
| X | 8 | 0 | 8 |
| Y | 7 | 4 | 11 |
| Z | 6 | 4 | 10 |
| Total | 31 | 10 | 41 |

## Sample selection

In Project 1, fecal samples originated from a convenience sample of five farms participating in a concurrent MAP prevalence study (in press). Samples were initially selected based on sufficient mass to allow division of the sample to provide adequate mass for each of the laboratories according to their specific requirements. Subsequent random allocation gave a total of 205 individual samples, grouped in 41 pools of five for analysis by each laboratory. A pool size of five was chosen as this is the maximum pool size laboratories B, C, and D offer as part of their commercial services. All five farms were sheep and beef mixed species farms. The sample set comprised 50 bovine and 155 ovine samples. The number of ovine and bovine pools tested from each farm is shown in Table 1.

Samples were collected between August and October 2021 by the first author. All samples were kept at 4°C and sent to the University of Liverpool within two to three days of being collected, where they were stored at −70°C. Individual samples were identified with a unique ID, and instructions on the individual pool composition were provided to each laboratory to ensure the same pre-defined pools were tested by each laboratory. Samples were shipped directly to laboratories B, C, and D from the University of Liverpool.

For Project 2, samples originated from a second, independent convenience sample of 190 ovine fecal samples collected from 10 flocks. These were pooled according to the same parameters as previously described into 38 pools of five individuals per pool and were analyzed by laboratories A and B only due to limited sample mass. Both projects were performed as blind studies.

## Laboratory assays

### Laboratory A

Samples were analyzed by the first author at the University of Liverpool research laboratories using the DNA extraction kit and RL-PCR assay developed by Kawaji et al. (20).

## Sample preparation

Individual samples were pooled into groups of five in the laboratory by creating individual fecal suspensions before pooling, as described by Mita et al. (21). Briefly, for each sample, 1 g of feces was placed into a 50-mL centrifuge tube containing 20 mL of sterile distilled water, mixed and shaken vigorously with a vortex for 30 sec before allowing the suspension to stand for 30 min, and repeated for all samples contributing to a pool of five. One milliliter of each fecal suspension was collected and pooled into a fresh 15-mL tube, followed by centrifugation at $900 \times g$ for 30 min at room temperature. All but approximately 1.5 mL of supernatant was removed, the fecal pellet was re-suspended in the remaining supernatant, and the entire volume was transferred to a tube containing zirconia beads provided with the Johne-PureSpin kit (FASMAC Ltd.).

## DNA extraction

DNA extraction was performed using the Johne-PureSpin kit (FASMAC Ltd.) according to the manufacturer's instructions. Briefly, 1 mL of supernatant from the prepared fecal suspension was transferred to the bead tube provided and centrifuged at 17,000 × $g$ for 6 min at room temperature. After removal of the supernatant, 400 µL of lysis buffer 1-A was added, and samples were pulverized for 20 min at 30 Hz using a tissue lyser (Qiagen Ltd.). Following centrifugation at 17,000 × $g$ for 6 min, the supernatant was transferred to a 1.5-mL tube containing 200 µL of lysis buffer 1-B and 75 µL of extraction buffer 2. After centrifugation at 17,000 × $g$ for 12 min, 500 µL of supernatant was mixed with 400 µL of binding buffer 3. The mixture was transferred to a spin column and centrifuged at 13,000 × $g$ for 1 min. The column was washed once with 600 µL of washing buffer 4, centrifuged at 13,000 × $g$ for 1 min, and then placed into a new 1.5-mL tube. The wash step was repeated if the sample was still discolored after the initial wash. DNA samples were eluted with 50 µL of elution buffer 5 by centrifugation at 13,000 × $g$ for 1 min and stored at −20°C if qPCR could not be performed the same day.

## qPCR assay

This was performed using a Reso-Light (RL) IS900 qPCR assay as described by Kawaji et al. (20). Briefly, the reaction mixture comprised 5 µL of template DNA, 25 µL of 2× GeneAce RL qPCR mix (Nippon Gene), 1 µL of 25 pmol of forward (IS900-3) and reverse (IS900-32) primers, 1 µL of 0.4fg/µL internal control (IC), 0.5 µL of 1 U/µL uracil-DNA glycosylase (UDG; Nippon Gene) and 16.5 µL of nuclease-free water to make up a 50 µL total reaction volume. Reaction mixtures were aliquoted with a QIAgility instrument (Qiagen Ltd.), and qPCR runs were performed on a Rotor-Gene Q machine (Qiagen Ltd.) on an SYBR reading setting using the following reaction program: initial incubation for UDG at 50°C for 2 min, followed by activation step of 10 min at 95°C, then 45 cycles of PCR amplification at 95°C for 30 secs and 68°C for 60 secs. After PCR amplification, dissociation curve data were collected to analyze melting temperature (Tm) peaks, as shown in Fig. 1. Tm peaks for the target (91.5°C ± 1.5°C) indicated the presence of MAP IS900, while the Tm peak for the IC (85.5°C ± 1.5°C) indicated a reaction without inhibition. If neither Tm peak was detected,

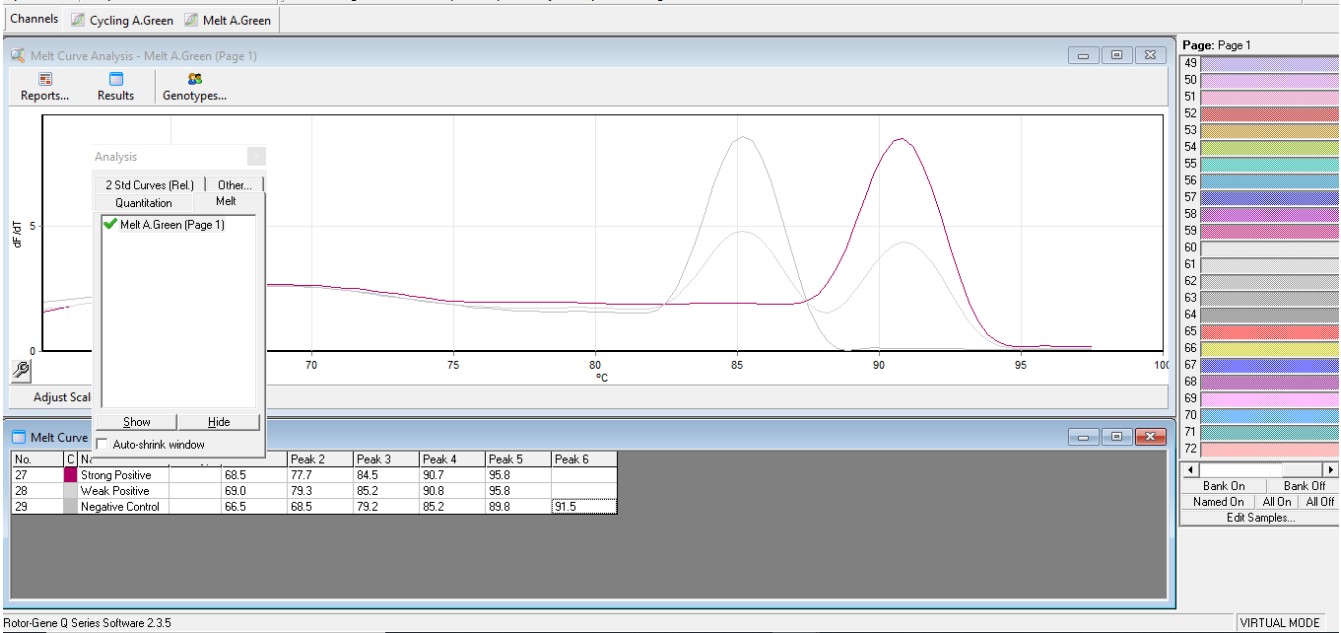

**FIG 1** Melt curve analysis for the Reso-Light (RL) IS900 qPCR assay used by laboratory A. Melting temperature peaks for the target IS900 and internal control and expected results for the strong and weak positive and negative controls.

the result was interpreted as invalid due to PCR inhibition. DNA extracts were run in singular with PCR-positive and negative controls and DNA extraction-negative controls.

## Laboratory B

Samples were analyzed according to laboratory B's specific protocols. The detailed specification of the protocol is not in the public domain. Based on personal communication with a laboratory representative, the author's understanding of the protocol is as follows: 2 g of fecal sample is required from each individual animal contributing to the pool of five. Fecal material is pooled, and 2 g of the composite fecal sample undergoes purification steps with water before centrifugation and resuspension in TE buffer in preparation for magnetic bead DNA extraction using a MagMax pathogen RNA/DNA kit (Applied Biosystems). qPCR is performed in duplicate for 45 cycles using the primer pair MP10-1 and MP11-11, designed to target the MAP IS900 element. Interpretation based on Ct values and melt curve value (Tm) is as follows: positive samples have a Ct value <45 combined with a Tm of between 87°C and 90°C inclusive. Negative samples have a Ct value >45 or <45 combined with an incorrect Tm product. Both replicates must be analyzed for all positive samples showing a Ct value of 35 or higher. If the replicate is also positive, it will be reported as such. If the replicate is negative, then it should be reported as negative. If there is still a discrepancy, e.g., one replicate has the correct Tm product and Ct value below 35 and the other replicate is clear negative, then, it is repeated a third time, and that result is reported. Repeats are done on the extracted DNA from the original extraction process. Samples are always reported as either positive or negative.

## Laboratory C

Samples were analyzed according to laboratory C's specific protocols. The detailed specification of the protocol is not in the public domain. Based on personal communication with a representative of the laboratory, the author's understanding of the protocol is as follows: 5 g of the fecal sample from each animal contributing to the pool of five is added to an EZ PREP bottle containing 30 mL of a proprietary buffer (IDVet) and then vortexed for 30 seconds. Two milliliter from each fecal suspension is then squeezed through a filter, located in the cap of the EZ Prep lid, into a container, and the resulting 10 mL is mixed by vortexing. Of the pooled suspension, 1.5 mL is added to the bead tube for magnetic bead DNA extraction using the ID Gene Mag Fast extraction kit (Innovative Diagnostics). qPCR is performed using the ID Gene Paratuberculosis Duplex kit (Innovative Diagnostics), which includes an internal control. DNA extracts are run in singular for 40 cycles. The interpretation is as follows: if Ct < 33, record the sample as positive, not re-tested. If Ct value > 40, record as negative. If Ct ≥ 33 and < 40, repeat from the beginning of the extraction process. If Ct < 40 on repeat, record the sample as positive. If Ct > 40 on repeat, record the sample as inconclusive. Statistical analyses were performed with inconclusive results coded as both positive (C1) and negative (C2) to assess interrater reliability.

## Laboratory D

Samples were analyzed according to laboratory D's specific protocols. The detailed specification of the protocol is not in the public domain. Based on personal communication with a representative of the laboratory, the author's understanding of the protocol is as follows: 1 g of feces from each animal contributing to the pool of five is mixed into 30 mL of water, of which 1 mL is used for DNA extraction using a MagMAX CORE Mechanical Lysis kit (Applied Biosystems). qPCR is performed using the VetMAX MAP Screening kit (Applied Biosystems). DNA extracts are run in singular for 45 cycles. The interpretation is as follows: if Ct < 37, record the sample as positive, not re-tested. If Ct value > 45, record as negative. Ct ≥ 37 and < 45, repeat from the beginning of the extraction process if sufficient amounts of original fecal sample remain, otherwise with saved material from part way through the extraction process. If Ct < 37 on repeat, record

as positive. If Ct ≥ 37 and < 45 on repeat, record as inconclusive or positive depending on clinical history. If Ct value > 45 on repeat, repeat extraction and record the best of three results.

## Statistical analysis

Interrater reliability was tested using Cohen's and Fleiss' kappa coefficients in Minitab19 Statistical Software (Minitab, LLC) to assess pairwise and overall agreement between all four laboratories, respectively, at the pool level.

## RESULTS

### Project 1

The numbers of positive pools for assays A, B, C1, C2, and D were 18, 12, 11, 2, and 1 (43.9%, 29.3%, 26.8%, 4.9%, and 2.4%), respectively. Figure 2 shows a heat map of the 41 raw results for each of the laboratories for farms V–Z for both ovine and bovine samples.

Figure 3 shows the results of each laboratory for ovine and bovine pools.

The Fleiss' kappa coefficient between all four laboratories for both bovine and ovine pools was 0.15 ($P$ = 0.011 and 0.008 for C1 and C2, respectively). Cohen's kappa scores ranged between −0.05 and 0.66 when considering all 41 samples, as shown in Table 2.

Fleiss' kappa coefficient values between all four laboratories for ovine samples were 0.19 ($P$ = 0.005 ) and 0.12 ($P$ = 0.050) for C1 and C2, respectively. Cohen's kappa scores ranged between −0.06 and 0.65 when considering all 31 samples, as shown in Table 3.

The Fleiss' kappa coefficient between all four laboratories for bovine samples was −0.10 ($P$ = 0.773) and 0.16 ($P$ = 0.109) for C1 and C2, respectively. Cohen's kappa scores ranged between −0.36 and 0.62 when considering all 10 samples, as shown in Table 4.

For the diagnosis of MAP infection at the farm level, laboratory A detected four out of the five flocks as MAP positive, laboratory B detected three, laboratory C detected two (for both interpretations of inconclusive results), and laboratory D detected only one flock as MAP positive, as shown in Fig. 4.

For diagnosis of MAP infection in the three beef herds in the study, laboratory A detected one herd as MAP positive, laboratories B and C1 detected two (but not the same two), while C2 and D did not detect any herds as MAP positive, as shown in Fig. 5.

### Project 2

The number of positive pools detected by laboratories A and B were 24 (63.1%) and 17 (44.7%), respectively, as shown in Table 5, with a Cohen's kappa score of 0.54. These results were similar to the ovine results of Project 1, whereby agreement between laboratories A and B was moderate. Furthermore, laboratory A's assay was again more sensitive than laboratory B's. When considering samples from Projects 1 and 2 together, Cohen's kappa scores for all ($n$ = 79) and ovine-only ($n$ = 69) samples were 0.57 and 0.54, respectively.

For diagnosis of MAP infection at the farm level, laboratory A diagnosed all 10 flocks as MAP positive, while laboratory B detected just six as MAP positive.

## DISCUSSION

Agreement between the four laboratories varied greatly and was generally remarkably poor for both ovine and bovine samples. Several kappa values below zero were recorded between laboratories B–D, representing agreement worse than expected by chance alone or disagreement. Low negative scores (0 to −0.10) may generally be interpreted as "no agreement," but large negative kappa values, such as −0.36, seen between laboratories B and C1 for bovine samples, represents substantial disagreement between these two operators. The highest interrater reliability was consistently seen between laboratories C2 and D, which was attributable to the high proportion of negative pooled results for both of these laboratories in both species. The most likely explanation for this

| Farm ID | Species | A | B | C1 | C2 | D |
|---------|---------|---|---|----|----|---|
| V | O | 0 | 0 | 0 | 0 | 0 |
| V | O | 1 | 0 | 0 | 0 | 0 |
| V | O | 0 | 0 | 0 | 0 | 0 |
| V | O | 0 | 0 | 0 | 0 | 0 |
| V | O | 0 | 0 | 0 | 0 | 0 |
| V | B | 0 | 1 | 0 | 0 | 0 |
| V | B | 0 | 0 | i | 0 | 0 |
| W | O | 0 | 0 | 0 | 0 | 0 |
| W | O | 0 | 0 | 0 | 0 | 0 |
| W | O | 0 | 0 | 0 | 0 | 0 |
| W | O | 0 | 0 | 0 | 0 | 0 |
| W | O | 0 | 0 | 0 | 0 | 0 |
| X | O | 1 | 0 | i | 0 | 0 |
| X | O | 1 | 0 | i | 0 | 0 |
| X | O | 1 | 1 | 0 | 0 | 0 |
| X | O | 1 | 1 | 0 | 0 | 0 |
| X | O | 1 | 1 | 0 | 0 | 0 |
| X | O | 1 | 1 | 1 | 1 | 0 |
| X | O | 1 | 1 | i | 0 | 0 |
| X | O | 1 | 1 | i | 0 | 0 |
| Y | O | 0 | 0 | 0 | 0 | 0 |
| Y | O | 0 | 0 | 0 | 0 | 0 |
| Y | B | 0 | 0 | i | 0 | 0 |
| Y | B | 0 | 0 | i | 0 | 0 |
| Y | O | 0 | 0 | 0 | 0 | 0 |
| Y | O | 0 | 0 | 0 | 0 | 0 |
| Y | O | 1 | 0 | 0 | 0 | 0 |
| Y | O | 0 | 0 | 0 | 0 | 0 |
| Y | O | 1 | 1 | 0 | 0 | 0 |
| Y | B | 0 | 0 | i | 0 | 0 |
| Y | B | 0 | 0 | 0 | 0 | 0 |
| Z | O | 1 | 0 | i | 0 | 0 |
| Z | O | 1 | 1 | 0 | 0 | 0 |
| Z | O | 1 | 0 | 1 | 1 | 1 |
| Z | O | 1 | 0 | 0 | 0 | 0 |
| Z | B | 0 | 0 | 0 | 0 | 0 |
| Z | B | 0 | 0 | 0 | 0 | 0 |
| Z | B | 0 | 0 | 0 | 0 | 0 |
| Z | B | 1 | 1 | 0 | 0 | 0 |
| Z | O | 1 | 1 | 0 | 0 | 0 |
| Z | O | 1 | 1 | 0 | 0 | 0 |

**FIG 2** Heat map of the raw pooled results for laboratories A, B, C1, C2, and D for farms V–Z for both ovine (O) and bovine (B) samples ($n = 41$). 1, positive; 0, negative; i, inconclusive.

| Farm ID | Total Number of Pools Tested | Laboratory | | | | |
|---|---|---|---|---|---|---|
| | | A | B | C1 | C2 | D |
| V | 5, 2 | 1, 0 | 0, 1 | 0, 1 | 0, 0 | 0, 0 |
| W | 5, 0 | 0, 0 | 0, 0 | 0, 0 | 0, 0 | 0, 0 |
| X | 8, 0 | 8, 0 | 6, 0 | 5, 0 | 1, 0 | 0, 0 |
| Y | 7, 4 | 2, 0 | 1, 0 | 0, 3 | 0, 0 | 0, 0 |
| Z | 6, 4 | 6, 1 | 3, 1 | 2, 0 | 1, 0 | 1, 0 |
| Grand Total | 31, 10 | 17, 1 | 10, 2 | 7, 4 | 2, 0 | 1, 0 |

FIG 3 Summary of number of positive ovine (black) and bovine (red) pools as detected by laboratories A, B, C1, C2, and D for farms V–Z.

agreement is the poor sensitivity of laboratory D and, to a slightly lesser extent, C2. These two results evoke a false sense of high interrater reliability and should be interpreted cautiously. Unfortunately, no "gold standard" diagnostic test was available to assess the "correctness" of the results in this ring trial. However, the high specificity performance demonstrated previously for the primer pairs employed by the assays performed by laboratories A (20) and B (17) confers a significant degree of confidence that positive results from these two assays are valid. For ovine samples, laboratory A detected the highest number of positive pools, followed by laboratory B. Agreement between these two sets of results was moderate (0.56 and 0.62 kappa values for ovine and bovine pools, respectively, for Project 1 and 0.54 for Project 2) and broadly in line with the 0.6 kappa score level considered to be the minimum acceptable interrater agreement for clinical and diagnostic settings (22). Laboratory C1 detected the highest number of positive pools for bovine samples, followed by laboratory B. However, the agreement between these two sets of results for bovine samples was remarkably poor, with a significant negative kappa value (−0.36). Furthermore, there wasn't sufficient fecal mass to assess repeatability by each laboratory, but this would have been another interesting test performance characteristic to investigate.

The marked difference in MAP detection rates at the flock and herd levels between the four laboratories highlights the clinical consequences of poor qPCR performance. Results from this ring trial indicate that the chance of diagnosing MAP infection in a flock is significantly affected by which laboratory is selected by submitting veterinary clinicians and the number of pools analyzed. This was also true for diagnosing MAP infection in beef herds, which is potentially even more concerning as laboratories B, C, and D all claim to have done significantly more internal validation and proficiency testing for MAP qPCR on bovine versus ovine feces. However, the relatively small sample size of bovine pools used in this ring trial makes it difficult to assess the significance of this observation.

TABLE 2 Agreement analysis between laboratories A, B, C1, C2, and D for all ovine and bovine samples ($n = 41$)

| Laboratory | A | B | C1 | C2 | D |
|---|---|---|---|---|---|
| A | X | 0.59[a](0)[b] | 0.22 (0.062) | 0.12 (0.051) | 0.06 (0.126) |
| B | | X | −0.03 (0.568) | 0.07 (0.254) | −0.05 (0.743) |
| C1 | | | X | X | 0.13 (0.047) |
| C2 | | | X | X | 0.66 (0) |
| D | | | | | X |

[a]Cohen's kappa statistic.
[b]$P$ (vs <0) of κ.

**TABLE 3** Agreement analysis between laboratories A, B, C1, C2, and D for ovine samples ($n = 31$)

| Laboratory | A | B | C1 | C2 | D |
|---|---|---|---|---|---|
| A | X | $0.56^a (0)^b$ | 0.39 (0.003) | 0.11 (0.092) | 0.05 (0.178) |
| B | | X | 0.12 (0.248) | 0.07 (0.290) | −0.06 (0.759) |
| C1 | | | X | X | 0.21 (0.030) |
| C2 | | | X | X | 0.65 (0) |
| D | | | | | X |

[a]Cohen's kappa statistic.
[b]$P$ (vs <0) of κ.

Some degree of disagreement between the four laboratories was expected, especially for samples containing low amounts of MAP DNA. MAP is known to "clump" in feces (17), and, for stochastic reasons, this non-homogenous distribution could explain a lower detection rate for samples with low MAP levels. All four laboratories use qualitative qPCR assays; thus, quantifying the bacterial load of the samples in this study was not possible to explore this hypothesis. The pooling process and resulting dilution factor would have exasperated low MAP DNA levels. It's also possible that MAP quantities varied between different sub-samples used by each laboratory from the same original individual animal, despite mixing the original sample before creating sub-samples to be sent to laboratories B–D.

It's also plausible that other factors may have contributed to the wide range of apparent sensitivities resulting in the poor levels of inter-laboratory agreement. Potential explanations for the wide variation in results between laboratories include differences in sample volume, pooling methodology, DNA extraction protocols, and primer pairs employed, all of which may affect sensitivity and interpretation of results.

Stipulated individual fecal sample volumes for assays A–D were 1, 2, 5, and 1 g, respectively. Interestingly, both the most and least sensitive assays (A and D, respectively) required the smallest volumes of feces to be submitted, which indicates variation in performance is not attributable to this factor alone. However, the low volume of feces required for assay D may have contributed to overall poor performance.

An appropriate DNA extraction kit is essential for maximizing the final eluted DNA concentrations and purity (23). Efficient extraction and DNA isolation are particularly challenging with a complex biological matrix such as feces and an organism such as MAP. Feces contain PCR-inhibiting substances such as polysaccharides, heme, plant components, phytic acid, polyphenols, bile salts, and Ca2+ ions, which can lead to false-negative results by inhibiting DNA amplification in the PCR (24), thus compromising the PCR reaction. Additionally, MAP bacteria possess a thick waxy cell wall with a high lipid content that renders them inherently difficult to be lysed, resulting in low DNA recovery and reduced PCR sensitivity (23, 25). Direct comparisons between the performance of the extraction kits used by laboratories A–D were not possible here, although it's feasible that this affected the overall diagnostic sensitivity.

The variation in sensitivity performance observed for laboratories A–D may also be due to the target sequence of each qPCR. Laboratories A, B, and C use primers to detect the insertion sequence IS900, which is considered a MAP-specific element with 15–20 copies present in a single MAP genome (26). However, IS900-like sequences

**TABLE 4** Agreement analysis between laboratories A, B C1, C2, and D for bovine samples ($n = 10$)

| Laboratory | A | B | C1 | C2 | D |
|---|---|---|---|---|---|
| A | X | $0.62^a (0.018)^b$ | −0.19 (0.805) | 0 (*) | 0(*) |
| B | | X | −0.36 (0.902) | 0 (*) | 0 (*) |
| C1 | | | X | X | 0(*) |
| C2 | | | X | X | N/A |
| D | | | | | X |

[a]Cohen's kappa statistic.
[b]$P$ (vs <0) of κ.

| Farm ID | Number of Pools | A | B | C1 | C2 | D |
|---------|-----------------|---|---|----|----|----|
| V | 5 | 1 | 0 | 0 | 0 | 0 |
| W | 5 | 0 | 0 | 0 | 0 | 0 |
| X | 8 | 8 | 6 | 5 | 1 | 0 |
| Y | 7 | 2 | 1 | 0 | 0 | 0 |
| Z | 6 | 6 | 3 | 2 | 1 | 1 |
| Total | 31 | 17 | 10 | 7 | 2 | 1 |

FIG 4   Number of positive ovine pools detected by each of laboratories A–D for farms V–Z. Highlighted cells show positive flock diagnoses.

do exist in non-MAP mycobacteria, which may contribute to false-positive results if the PCR method employs common probes and primers (27). This can be overcome by designing and employing specific primer sets. The study which first described the primer pair (MP10-1 and MP11-1) used by laboratory B reported no cross-reaction when conducting specificity experiments using 51 non-MAP mycobacterial species, including 10 that contained IS900-like sequences (17). Similarly, the study which first described the primer pair (IS900-3 and IS900-32) used by laboratory A also reported 100% specificity when evaluating against 57 other non-MAP mycobacterial strains (20). Furthermore, laboratories A and B used assessment of melting temperatures as part of their interpretation, providing further reassurance regarding specificity. Laboratory D's target gene is ISMAP02 which has a lower copy number (six copies) in the MAP genome (28) than IS900, which may explain the lower diagnostic sensitivity.

Differences in pooling methodology may also have affected laboratory performance. To ensure all individuals in a pool are represented in the sample that is used for DNA extraction, it is mandatory that pooled samples are thoroughly mixed, in particular, to mitigate MAP's clumping tendencies. It is arguably more difficult to effectively mix solid feces (as for the protocols for laboratories B and D) versus liquid fecal suspensions (laboratories A and C).

The MAP qPCR assays offered by two of the four laboratories involved in this ring trial study are accredited by the United Kingdom Accreditation Service (UKAS) and Cattle Health Certification Standards (CHECS) for bovine fecal samples and can be used interchangeably by farmers enrolled in CHECS-licensed herd health schemes. CHECS is the body that sets standards, quality controls, and certifies the official JD control programs in GB. This ring trial focused on assessing the performance of MAP qPCRs as a screening test on pooled samples, while these control programs most commonly employ MAP qPCR assays for individual bovine samples. However, it's plausible that the poor performance of a particular laboratory on pooled samples also translates to poor performance on individual samples, particularly for low-shedding animals. If this is the case, the results of this ring trial would suggest that the extremely poor sensitivity of assay D raises the question of whether it is fit for purpose as an approved diagnostic test for such health schemes. Indeed, anecdotal reports from cattle farmers and veterinary practitioners raising concerns over the divergent performance of MAP qPCR assays

| Farm ID | Number of Pools | A | B | C1 | C2 | D |
|---------|-----------------|---|---|----|----|----|
| V | 2 | 0 | 1 | 1 | 0 | 0 |
| Y | 4 | 0 | 0 | 3 | 0 | 0 |
| Z | 4 | 1 | 1 | 0 | 0 | 0 |
| Total | 10 | 1 | 2 | 4 | 0 | 0 |

FIG 5   Number of positive bovine pools detected by each of laboratories A–D for farms V–Z. Highlighted cells show positive herd diagnoses.

**TABLE 5** Summary of number of positive pools as detected by laboratories A and B for farms A–J

| Farm | Number of pools tested | Number of positive pools | |
|---|---|---|---|
| | | Laboratory A | Laboratory B |
| A | 7 | 7 | 6 |
| B | 3 | 2 | 1 |
| C | 3 | 3 | 3 |
| D | 5 | 4 | 5 |
| E | 1 | 1 | 1 |
| F | 3 | 1 | 0 |
| G | 4 | 2 | 1 |
| H | 2 | 2 | 0 |
| I | 4 | 1 | 0 |
| J | 6 | 1 | 0 |
| Total | 38 | 24 | 17 |

from laboratories C and D are not uncommon. The inability to diagnose MAP-positive individuals leads to unintentionally retaining infected animals and their progeny within the herd, maintaining a reservoir of infection that will propagate disease spread. This thwarts costly efforts made at JD control and decreases consumer confidence. Perversely, laboratory B demonstrated the highest diagnostic sensitivity of the three commercially available assays assessed in this ring trial, and yet, the MAP qPCR assay offered by laboratory B is neither UKAS nor CHECS accredited.

The dilution effects of pooling samples may have reduced the sensitivity of the assays assessed in this ring trial. However, commercial laboratories should offer a qPCR assay capable of detecting low amounts of MAP DNA in pooled samples as a cost-effective screening option for livestock producers. The pool size of five used in this study is the maximum pool size laboratories B–D will run, and yet, other countries routinely offer MAP diagnostic tests in larger pool sizes. The Australian Johne's Disease Market Assurance Program for Sheep (SheepMAP) uses pool sizes of 50 animals for pooled fecal culture (29), and there a several reports in the literature of pool sizes of 10 having no adverse effect on qPCR sensitivity compared with individual testing (20, 30).

## Conclusion

Variation between commercial laboratories offering MAP qPCR assays is a significant concern. The three main MAP qPCR assays currently available commercially in GB appear to have inferior sensitivity to other DNA extraction and qPCR kit combinations described in the literature and commercially available on the broader global market. A lack of diagnostic sensitivity will lead to the underdiagnosis of this important disease and subsequently a loss of consumer confidence in the laboratory tests and the health schemes that endorse them. Further work is required to standardize the performance of assays within and between laboratories for both bovine and ovine samples, with specific validation and proficiency studies undertaken for each species. This would assist farmers and veterinarians in making more informed decisions on screening, diagnosis, and control of Johne's disease on GB livestock farms.

## ACKNOWLEDGMENTS

Funding was provided by Hybu Cig Cymru, Virbac Ltd., and the University of Liverpool.

## AUTHOR AFFILIATION

[1]Department of Livestock and One Health, Institute of Infection, Veterinary and Ecological Sciences, University of Liverpool, Leahurst Campus, Neston, Cheshire, United Kingdom

## AUTHOR ORCIDs

L. Worsley ⓘ http://orcid.org/0000-0002-1024-3511

## FUNDING

| Funder | Grant(s) | Author(s) |
|---|---|---|
| Virbac Ltd | | P. L. Davies |
| Hybu Cig Cymru | | P. L. Davies |
| University of Liverpool (UoL) | | P. L. Davies |

## ADDITIONAL FILES

The following material is available online.

Open Peer Review

**PEER REVIEW HISTORY (review-history.pdf).** An accounting of the reviewer comments and feedback.

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
