## [Reviewer comments · Microbiology Spectrum]

Microbiology Spectrum

Inter-laboratory ring trial to compare four quantitative polymerase chain reaction assays employed for detection of *Mycobacterium avium* subspecies paratuberculosis

Laura Worsley and Peers Davies

Corresponding Author(s): Laura Worsley, University of Liverpool

Review Timeline:

Submission Date:	May 26, 2023
Editorial Decision:	August 14, 2023
Revision Received:	September 19, 2023
Editorial Decision:	October 12, 2023
Revision Received:	December 4, 2023
Accepted:	December 18, 2023

Editor: Sadjia Bekal

Reviewer(s): Disclosure of reviewer identity is with reference to reviewer comments included in decision letter(s). The following individuals involved in review of your submission have agreed to reveal their identity: Fazeela Arshad (Reviewer #2); Asra'a Abdul-Jalil (Reviewer #3); Muthuraj Muthaiah (Reviewer #4)

Transaction Report:

DOI: <https://doi.org/10.1128/spectrum.02210-23>

July 31, 2023

Dr. Laura Worsley
University of Liverpool
Chester High Road
Neston CH64 7TE
United Kingdom

Re: Spectrum02210-23 (Inter-laboratory ring trial to compare four quantitative polymerase chain reaction assays employed for detection of *Mycobacterium avium* subspecies paratuberculosis)

Dear Dr. Laura Worsley:

Link Not Available

Sincerely,

Sadjia Bekal

Journals Department
Reviewer comments:

Reviewer #2 (Comments for the Author):

Please consider including the qPCR analysis curves, as they would provide visual evidence of the positivity of the reaction.

Reviewer #3 (Comments for the Author):

1. At the end of line 14, What's number(1) mean? is it a reference number? The abstract mustn't contain any citations.

2. In line 15, GB is short for what, usually in the abstract doesn't write an abbreviation without the full name.
3. LW.....shorts for what?
4. et.al must be written in italic.
5. The Author should follow a unique style in reference citations, a number of references without year.
6. (the author's understanding of the protocol) In line 177, is the author create this protocol or retrieve it from other literature? can you illustrate this point?
7. The writing language had multi grammar mistakes, especially in tenses.
8. In materials & methods, the target(specific gene or a region in the genome) in qPCR isn't clear also housekeeping genes.

Reviewer #4 (Comments for the Author):

Need major revision

Staff Comments:

Preparing Revision Guidelines

Please return the manuscript within 60 days; if you cannot complete the modification within this time period, please contact me. If you do not wish to modify the manuscript and prefer to submit it to another journal, please notify me of your decision immediately so that the manuscript may be formally withdrawn from consideration by Microbiology Spectrum.

Inter-laboratory ring trial to compare four quantitative polymerase chain reaction assays employed for detection of *Mycobacterium avium* subspecies *paratuberculosis*

Authors: Worsley L^{1*}, Davies PL¹

Author affiliations: ¹ Department of Livestock & One Health, Institute of Infection Veterinary & Ecological Sciences, University of Liverpool, Leahurst Campus, Cheshire CH64 7TE.

*corresponding author: L.Taylor11@liverpool.co.uk

Abstract

Johne's disease is an infectious enteric disease caused by *Mycobacterium avium* subspecies *paratuberculosis* (MAP) affecting ruminant species worldwide. In Project 1, an independent performance comparison ring trail was conducted between three different commercial MAP quantitative polymerase chain reaction (qPCR) assay services (B, C and D) currently marketed in Great Britain by three separate laboratories against each other and against a fourth assay (A) (1) not currently available commercially in GB. A total of 205 individual ovine and bovine samples from 5 farms were analysed to give 41 sets of pooled results (pool size 5) from each laboratory according to their own specific protocols. The number of positive pools for assays A-D were 18, 12, 11 and 1 (43.9, 29.2, 26.8 and 2.4%) respectively. Assessment of interrater reliability resulted in a Fleiss' kappa coefficient of 0.15 indicating very poor overall agreement between the 4 laboratories. At the farm level, laboratories A-D diagnosed 4, 3, 2 and 1 flocks respectively as MAP positive. In Project 2, 38 pooled ovine samples from 10 flocks were analysed to compare performance of laboratories A and B. The number of positive results for laboratories A and B were 24 (63.1%) and 17 (44.7%) respectively (Cohen's kappa 0.54), indicating that laboratory A was more sensitive than B in line with results from Project 1. Variation between laboratories offering MAP qPCR assays is a significant concern and further work is warranted to validate and standardise the performance of assays between laboratories for both ovine and bovine samples.

Importance

Our study reports the findings of an inter-laboratory ring trail comparing the performance of four different qPCR assay services for the detection of *Mycobacterium avium* subspecies *paratuberculosis* (MAP) infection in cattle and sheep. MAP is the causative agent of Johne's disease (also known as paratuberculosis), a significant production-limiting disease in livestock populations with a worldwide distribution. The content of this paper is significant and novel as it is the first to highlight the marked variation between the diagnostic sensitivity and reproducibility of the three principal commercial laboratories offering MAP qPCR diagnostic and screening services in Great Britain. The low sensitivity and high degree of variability between the laboratories is of great concern and relevance to veterinary practitioners and livestock producers.

INTRODUCTION

Johne's disease (JD), or paratuberculosis, is an important production-limiting chronic wasting disease caused by *Mycobacterium avium* subspecies *paratuberculosis* (MAP) affecting predominantly ruminants with a worldwide distribution. The disease is chronic and insidious in nature with a long incubation period during which infected animals shed MAP in their faeces, intermittently and at low

levels initially (2, 3) progressing to persistent and heavy shedding as the disease progresses towards the clinical stages for those animals that are not capable of clearing the infection (4, 5). In addition to negatively impacting animal welfare, JD is responsible for economic losses through reduced productivity, increased replacement costs associated with premature culling and increased mortality (6, 7). To this end, disease control programmes have been developed in many countries including Great Britain (8). These programmes require robust and reliable diagnostic and screening tests to establish herd or flock MAP infection status, to identify infected individual animals for culling and make informed decisions on retention of their progeny, identify uninfected herds/ flocks that are free from disease as a source of replacement stock and also to satisfy specific health scheme requirements. Paratuberculosis also raises public health concerns with ongoing contention around MAP's suspected role in the development of Crohn's disease in humans. This places additional emphasis on the importance of implementing on-farm JD control programmes (9).

Antemortem diagnosis of MAP infection in the early sub-clinical stages of disease is challenging due to the lack of sufficiently sensitive and cost-effective tests. Additionally, whilst commercially available MAP qPCR assays are validated for bovine samples in GB, the same laboratories do not offer validated equivalent tests for ovine samples. The general consensus is that faecal qPCR and culture are more sensitive than antibody tests especially in the early stages of disease as MAP faecal shedding precedes a detectable antibody response in sheep and cattle (10-12).

Antibody ELISA tests are most appropriate in the more advanced stages of disease associated with increasingly discernible clinical signs (2). However, despite their low sensitivity in the early stages of infection, ELISA assays are frequently employed as herd and flock screening tests because of their practicality and low cost. Indeed, whole herd serological screening using ELISA tests are routinely used for official JD control programmes for GB beef herds and also for sheep flocks albeit much less commonly.

Whilst faecal culture is still often considered 'gold standard' largely due to its very high specificity (13) sensitivity estimates vary considerably dependant on the MAP strain in question and stage of infection and thus magnitude of faecal shedding in the individual or population being tested (14, 15). In contrast to C-strains which are relatively amenable to laboratory culture techniques, S-strains display slow and fastidious growth and are highly sensitive to various decontamination procedures and certain antibiotics (16, 17). As a result, it's plausible that the sensitivity of faecal culture is underestimated where S strains predominate (15). How relevant this is to GB is unknown, given that there is a paucity of information regarding circulating MAP strains in our sheep and cattle populations. Faecal real-time qPCR assays have been shown to demonstrate superior sensitivity to faecal culture (15, 18-20). Plausible explanations for this include the ability of qPCR to detect both C- and S- strains and also to detect live as well as dead MAP organisms (15).

To the author's knowledge, there have been no published independent trials comparing the performance of the different MAP qPCR assays commercially available in GB. The aim of this study was to establish the comparable performance of commercially available pooled faecal MAP qPCR services that are available as screening tests to veterinarians in clinical practice. The inter-laboratory ring trial study design encompasses the combined effect of intrinsic biochemical and extrinsic procedural factors that influence the sensitivity and specificity of the testing service provided to the clinician.

MATERIALS AND METHODS

Two projects were conducted as part of this inter-laboratory ring trial study. Project 1 compared the performance of 3 pooled faecal MAP qPCR tests (assay services B, C and D) as provided by the 3 prominent laboratories (B, C and D respectively) offering this assay in GB against each other and also

against a fourth PCR (A) (1) not currently available commercially in GB and performed at University of Liverpool (laboratory A). Laboratories B and C are government subsidised whilst laboratory D is a private commercial enterprise. Project 2, compared the performance of laboratory A against laboratory B using a further set of samples.

Sample Selection

In Project 1, faecal samples originated from a convenience sample of 5 farms participating in a concurrent MAP prevalence study (in press). Samples were initially selected based on sufficient mass to allow division of the sample to provide adequate mass for each of the laboratories according to their specific requirements. Random allocation thereafter gave a total of 205 individual samples, grouped in 41 pools of 5 for analysis by each laboratory. A pool size of 5 was chosen as this is the maximum pool size laboratories B, C and D offer as part of their commercial services. All 5 farms were sheep and beef mixed species farms. The sample set comprised 50 bovine and 155 ovine samples. The number of ovine and bovine pools tested from each farm are shown in Table 1.

Table 1. Summary of total number of ovine and bovine pools analysed from farms V-Z.

Farm ID	Number of Ovine Pools Tested	Number of Bovine Pools Tested	Total Number of Pools Tested
V	5	2	7
W	5	0	5
X	8	0	8
Y	7	4	11
Z	6	4	10
Total	31	10	41

Samples were collected between August and October 2021 by LW. All samples were kept at 4°C and sent to the University of Liverpool within 2-3 days of being collected where they were stored at -70°C. Individual samples were identified with a unique id and instructions on the individual pool composition were provided to each laboratory to ensure the same pre-defined pools were tested by each laboratory. Samples were shipped directly to laboratories B, C and D from University of Liverpool.

For Project 2, samples originated from a second, independent convenience sample of 190 ovine faecal samples collected from 10 flocks. These were pooled according to the same parameters as previously described into 38 pools of 5 individuals per pool and were analysed by laboratory A and B only due to limited sample mass. Both projects were performed as blind studies.

Laboratory Assays

Laboratory A

Samples were analysed by LW at the University of Liverpool research laboratories using the DNA extraction kit and RL-PCR assay developed by Kawaji et al (1).

Sample Preparation. Individual samples pooled into groups of 5 in the laboratory by creating individual faecal suspensions prior to pooling as described by Mita et al (21). Briefly, for each

individual sample, 1g of faeces placed into a 50ml centrifuge tube containing 20ml of sterile distilled water, mixed and shaken vigorously with a vortex for 30 sec before allowing the suspension to stand for 30 min. Repeated for all samples contributing to a pool of 5. One ml of each faecal suspension subsequently collected and pooled into a fresh 15ml tube followed by centrifugation at 900 g for 30 min at room temperature. All but approximately 1.5 ml of supernatant removed, faecal pellet re-suspended in remaining supernatant and entire volume transferred to tube containing zirconia beads provided with Johne-PureSpin kit (FASMAC Ltd).

DNA Extraction. DNA extraction performed using the Johne-PureSpin kit (FASMAC Ltd), according to the manufacturer's instructions. Briefly, 1ml of supernatant from prepared faecal suspension transferred to bead tube provided and centrifuged at 17k g for 6 min at room temperature. After removal of the supernatant, 400µl of lysis buffer 1-A added and samples pulverised for 20 min at 30Hz using a tissue lyser (Qiagen Ltd). Following centrifugation at 17K g for 6 min, supernatant transferred to 1.5ml tube containing 200µl of lysis buffer 1-B and 75µl of extraction buffer 2. After centrifugation at 17K g for 12 min, 500µl of supernatant mixed with 400µl of binding buffer 3. Mixture transferred to spin column and centrifuged at 13Kg for 1 min. Column washed once with 600µl of washing buffer 4, centrifuged at 13K g for 1 min, then placed into new 1.5ml tube. Wash step repeated if sample still discoloured after initial wash. DNA samples eluted with 50µl of elution buffer 5 by centrifugation at 13K g for 1 min and stored at -20°C if PCR could not be performed same day.

PCR assay. Performed using Reso-Light (RL) IS900 qPCR assay as described by Kawaji et al (1). Briefly, reaction mixture comprised 5µl of template DNA, 25µl of 2x GeneAce RL qPCR mix (Nippon Gene), 1µl of 25pmol of forward (IS900-3) and reverse (IS900-32) primers, 1µl of 0.4fg/µl internal control (IC), and 0.5µl of 1U/µl uracil-DNA glycosylase (UDG; Nippon Gene) and 16.5µl of nuclease-free water to make up 50µl total reaction volume. Reaction mixtures aliquoted with QIAgility instrument (Qiagen Ltd) and qPCR runs performed on Rotor-Gene Q machine (Qiagen Ltd) on SYBR reading setting using the following reaction program: initial incubation for UDG at 50°C for 2 mins, followed by activation step of 10 mins at 95°C, then 45 cycles of PCR amplification at 95°C for 30 secs and 68°C for 60 secs. After PCR amplification, dissociation curve data collected for analysing melting temperature (T_m) peaks. T_m peaks for the target (91.5 ± 1.5°C) indicated presence of MAP IS900, whilst T_m peak for the IC (85.5 ± 1.5°C) indicated a reaction without inhibition. If neither T_m peak detected, result interpreted as invalid due to PCR inhibition. DNA extracts ran in singular with PCR positive and negative controls and DNA extraction negative controls.

Laboratory B

Samples were analysed according to laboratory B's specific protocols. Detailed specification of the protocol is not in the public domain. Based on personal communication with a representative of the laboratory, the author's understanding of the protocol is as follows: 2g of faecal sample required from each individual animal contributing to the pool of 5. Faecal material is pooled and 2g of composite faecal sample undergoes purification steps with water before centrifugation and resuspension in TE buffer in preparation for magnetic bead DNA extraction using MagMax pathogen RNA/DNA kit (Applied Biosystems). PCR performed in duplicate for 45 cycles using the primer pair MP10-1 and MP11-11 that were designed to target the MAP IS900 element³. Interpretation based on Ct values and melt curve value (T_m) are as follows: Positive samples have a Ct value combined with a T_m of between 87-90°C inclusive. Negative samples either have a Ct value >45 or a Ct value <45 combined with an incorrect T_m product. For all positive samples showing a Ct value of 35 or higher both replicates need to be analysed together. If the replicate is also positive it will be reported as such. If the replicate is negative, then it should be reported as negative. If there is still a discrepancy e.g. 1 replicate has the correct T_m product and Ct value below 35 and the other replicate is clear negative,

then it is repeated a third time and that result is reported. Repeats are done on the extracted DNA from the original extraction process. Samples are always reported as either positive or negative.

Laboratory C

Samples were analysed according to laboratory C's specific protocols. Detailed specification of the protocol is not in the public domain. Based on personal communication with a representative of the laboratory, the author's understanding of the protocol is as follows: 5g of faecal sample from each individual animal contributing to the pool of 5 is added to an EZ PREP bottle containing 30 ml of a proprietary buffer (IDVet), then vortexed for 30 seconds. 2 ml from each individual faecal suspension is then squeezed through a filter, located in the cap of the EZ Prep lid, into a container and the resulting 10ml mixed by vortexing. 1.5 ml of the pooled suspension is added to the beads tube for magnetic bead DNA extraction performed using the ID Gene Mag Fast extraction kit (Innovative Diagnostics). PCR performed using the ID Gene Paratuberculosis Duplex kit (Innovative Diagnostics) which includes an internal control. DNA extracts are run in singular for 40 cycles. Interpretation is as follows: Ct < 33, record sample as positive, not re-tested. Ct value >40, record as negative. Ct \geq 33, repeat from the beginning of the extraction process. If Ct \geq 33 on repeat, record sample as positive. If Ct value >40 on repeat, record sample as inconclusive. For comparison to the other laboratories, statistical analyses were performed in parallel with inconclusive results coded as both positive (C1) and negative (C2).

Laboratory D

Samples were analysed according to laboratory D's specific protocols. Detailed specification of the protocol is not in the public domain. Based on personal communication with a representative of the laboratory, the author's understanding of the protocol is as follows: 1g of faeces from each individual animal contributing to the pool of 5 is mixed into 30ml of water, of which 1ml is used for DNA extraction using MagMAX CORE Mechanical Lysis kit (Applied Biosystems). PCR performed using the VetMAX MAP Screening kit (Applied Biosystems). DNA extracts are run in singular for 45 cycles. Interpretation is as follows: Ct < 37, record sample as positive, not re-tested. Ct value >45, record as negative. Ct \geq 37, repeat from the beginning of the extraction process if sufficient amounts of original faecal sample remains, otherwise with saved material from part way through the extraction process. If Ct < 37 on repeat, record as positive. If Ct \geq 37 on repeat, record as inconclusive or positive depending on clinical history. If Ct value >45 on repeat, repeat extraction and record best of 3 results.

Statistical analysis

Interrater reliability was tested using Cohen's and Fleiss' kappa coefficients in Minitab®19 Statistical Software (Minitab, LLC) to assess pairwise and overall agreement between all 4 laboratories respectively at the pool level.

RESULTS

Project 1

The number of positive pools for assays A, B, C1, C2 and D were 18, 12, 11, 2 and 1 (43.9, 29.3, 26.8, 4.9 and 2.4%) respectively. Table 2 shows a heat map of the 41 raw results for each of the laboratories for farms V-Z for both ovine and bovine samples.

Table 2 Heat map of the raw pooled results for laboratories A, B, C1, C2 and D for farms V-Z for both ovine (O) and B (bovine) samples (n=41). 1 = positive, 0 = negative, i = inconclusive.

Farm ID	Species	A	B	C1	C2	D
V	O	0	0	0	0	0
V	O	1	0	0	0	0
V	O	0	0	0	0	0
V	O	0	0	0	0	0
V	O	0	0	0	0	0
V	B	0	1	0	0	0
V	B	0	0	i	0	0
W	O	0	0	0	0	0
W	O	0	0	0	0	0
W	O	0	0	0	0	0
W	O	0	0	0	0	0
W	O	0	0	0	0	0
W	O	0	0	0	0	0
W	O	0	0	0	0	0
W	O	0	0	0	0	0
W	O	0	0	0	0	0
W	O	0	0	0	0	0
X	O	1	0	i	0	0
X	O	1	0	i	0	0
X	O	1	1	0	0	0
X	O	1	1	0	0	0
X	O	1	1	0	0	0
X	O	1	1	0	0	0
X	O	1	1	1	1	0
X	O	1	1	i	0	0
X	O	1	1	i	0	0
Y	O	0	0	0	0	0
Y	O	0	0	0	0	0
Y	B	0	0	i	0	0
Y	B	0	0	i	0	0
Y	O	0	0	0	0	0
Y	O	0	0	0	0	0
Y	O	0	0	0	0	0
Y	O	1	0	0	0	0
Y	O	0	0	0	0	0
Y	O	1	1	0	0	0
Y	B	0	0	i	0	0
Y	B	0	0	0	0	0
Z	O	1	0	i	0	0
Z	O	1	1	0	0	0
Z	O	1	0	1	1	1
Z	O	1	0	0	0	0
Z	B	0	0	0	0	0
Z	B	0	0	0	0	0
Z	B	0	0	0	0	0
Z	B	1	1	0	0	0
Z	O	1	1	0	0	0
Z	O	1	1	0	0	0

Tables 3 shows the results of each laboratory for ovine and bovine pools.

Table 3 Summary of number of positive ovine (black) and bovine (red) pools as detected by laboratories A, B, C1, C2 and D for farms V-Z.

Farm ID	Total Number of Pools Tested	Laboratory				
		A	B	C1	C2	D
V	5, 2	1, 0	0, 1	0, 1	0, 0	0, 0
W	5, 0	0, 0	0, 0	0, 0	0, 0	0, 0
X	8, 0	8, 0	6, 0	5, 0	1, 0	0, 0
Y	7, 4	2, 0	1, 0	0, 3	0, 0	0, 0
Z	6, 4	6, 1	3, 1	2, 0	1, 0	1, 0
Grand Total	31, 10	17, 1	10, 2	7, 4	2, 0	1, 0

The Fleiss' kappa coefficient between all four laboratories for both bovine and ovine pools was 0.15 ($p = 0.011$ and 0.008 for C1 and C2 respectively). Cohen's kappa scores ranged between -0.05 and 0.66 when considering all 41 samples as shown in Table 4.

Table 4 Agreement analysis between laboratories A, B C1, C2 and D for all ovine and bovine samples ($n=41$).

Laboratory	A	B	C1	C2	D
A		$0.59^1(0)^2$	0.22 (0.062)	0.12(0.051)	0.06 (0.126)
B			-0.03(0.568)	0.07 (0.254)	-0.05 (0.743)
C1					0.13(0.047)
C2					0.66 (0)
D					

1. Cohen's kappa statistic
2. p ($vs < 0$) of κ

The Fleiss' kappa coefficient between all four laboratories for ovine samples was 0.19 and 0.12 ($p = 0.005$ and 0.0501) for C1 and C2 respectively. Cohen's kappa scores ranged between -0.06 and 0.65 when considering all 31 samples, as shown in Table 5.

Table 5 Agreement analysis between laboratories A, B C1, C2 and D for ovine samples ($n=31$).

Laboratory	A	B	C1	C2	D
A		$0.56^1(0)^2$	0.39 (0.003)	0.11(0.092)	0.05(0.178)
B			0.12 (0.248)	0.07 (0.290)	-0.06 (0.759)
C1					0.21(0.030)

C2					0.65 (0)
D					

1. Cohen's kappa statistic
2. p (vs < 0) of κ

The Fleiss' kappa coefficient between all four laboratories for bovine samples was -0.10 ($p = 0.773$) and 0.16 ($p = 0.109$) for C1 and C2 respectively. Cohen's kappa scores ranged between -0.36 and 0.62 when considering all 10 samples, as shown in Table 6.

Table 6 Agreement analysis between laboratories A, B, C1, C2 and D for bovine samples ($n=10$).

Laboratory	A	B	C1	C2	D
A		0.62 ¹ (0.018) ²	-0.19 (0.805)	0 (*)	0(*)
B			-0.36 (0.902)	0 (*)	0 (*)
C1					0(*)
C2					N/A
D					

1. Cohen's kappa statistic
2. p (vs < 0) of κ

For diagnosis of MAP infection at the farm level, laboratory A detected 4 out of the 5 flocks as MAP positive, laboratory B detected 3, laboratory C detected 2 (for both interpretations of inconclusive results) and laboratory D only 1 flock as MAP positive as shown in Table 7.

Table 7 Number of positive ovine pools detected by each of Laboratories A-D for farms V-Z. Highlighted cells show positive flock diagnoses.

Farm ID	Number of Pools	A	B	C1	C2	D
V	5	1	0	0	0	0
W	5	0	0	0	0	0
X	8	8	6	5	1	0
Y	7	2	1	0	0	0
Z	6	6	3	2	1	1
Total	31	17	10	7	2	1

For diagnosis of MAP infection on the three beef herds in the study, laboratory A detected 1 herd as MAP positive, laboratories B and C1 detected 2 (but not the same two) whilst C2 and D did not detect any herds as MAP positive as shown in Table 8.

Table 8 Number of positive bovine pools detected by each of Laboratories A-D for farms V-Z. Highlighted cells show positive herd diagnoses.

Farm ID	Number of Pools	A	B	C1	C2	D
V	2	0	1	1	0	0

Y	4	0	0	3	0	0
Z	4	1	1	0	0	0
Total	10	1	2	4	0	0

Project 2

The number of positive pools detected by laboratories A and B were 24 (63.1%) and 17 (44.7%) respectively with a Cohen's kappa score of 0.54. These results were similar to the ovine results of Project 1 whereby agreement between laboratories A and B was moderate. Further, laboratory A's assay was again more sensitive than laboratory B's assay. When considering samples from both project 1 and 2 together, the Cohen's kappa score for all (n=79) and ovine only samples (n=69) was 0.57 for and 0.54 respectively.

For diagnosis of MAP infection at the farm level, laboratory A diagnosed all 10 flocks as MAP positive whilst laboratory B detected just 6 flocks as MAP positive.

Table 9 Summary of number of positive pools as detected by laboratories A and B for farms A-J.

Farm	Number of pools tested	Number of positive pools	
		Laboratory A	Laboratory B
A	7	7	6
B	3	2	1
C	3	3	3
D	5	4	5
E	1	1	1
F	3	1	0
G	4	2	1
H	2	2	0
I	4	1	0
J	6	1	0
Total	38	24	17

Discussion

Agreement between the four laboratories varied greatly and was generally remarkably poor for both ovine and bovine samples. Several kappa values below zero were recorded between laboratories B-D, which represents agreement worse than expected by chance alone, or disagreement. Low negative scores (0 to -0.10) may generally be interpreted as "no agreement" but large negative kappa values, such as -0.36 seen between laboratories B and C1 for bovine samples, represents substantial disagreement between these two operators. The highest interrater reliability was consistently seen between laboratories C2 and D and this was attributable to the high proportion of negative pooled results for both of these laboratories in both species. The most likely explanation for this agreement is poor sensitivity of laboratory D and to a slightly lesser extent C2. These two sets of results evoke a false sense of high interrater reliability and should be interpreted with caution. Unfortunately, there was no 'gold standard' diagnostic test available to assess 'correctness' of the results in this ring trial. However, the high specificity performance demonstrated previously for the primer pairs employed by the assays performed by laboratories A (1) and B (18) confers a significant degree of confidence that positive results from these two assays are valid. For ovine samples, laboratory A detected the highest number of positive pools followed by laboratory B. Agreement between these two sets of results was moderate (0.56 and 0.62 kappa values for ovine and bovine pools respectively for Project 1, and 0.54

for Project 2) and broadly in line with the 0.6 kappa score level considered to be the minimum acceptable interrater agreement for clinical and diagnostic settings (22). For bovine samples, laboratory C1 detected the highest number of positive pools followed by laboratory B. However, the agreement between these two sets of results for bovine samples was remarkably poor with a significant negative kappa value (-0.36). Further, there wasn't sufficient faecal mass to be able to assess repeatability by each laboratory but this would have been another interesting test performance characteristic to investigate.

The clinical consequences of poor qPCR performance are highlighted in the marked difference in detection rate of MAP infection at the flock and herd level by the different laboratories. Results from this ring trial indicate that the chance of diagnosing MAP infection in a flock is significantly affected by which laboratory is selected by submitting veterinary clinicians as well as the number of pools analysed. This was also true for diagnosing MAP infection in beef herds which is potentially even more concerning as laboratories B, C and D all claim to have done significantly more internal validation and proficiency testing for MAP PCR on bovine versus ovine faeces. However, the relatively small sample size of bovine pools used in this ring trial make it difficult to assess the significance of this observation.

Some degree of disagreement between the four laboratories was to be expected especially for samples that contained low amounts of MAP DNA. MAP is known to 'clump' in faeces (18) and, for stochastic reasons, this non-homogenous distribution could explain a lower detection rate for any samples with low MAP levels. All of the four laboratories use qualitative qPCR assays and thus quantifying bacterial load of the samples in this study was not possible to explore this hypothesis. Low MAP DNA levels would have been exasperated by the pooling process and resulting dilution factor. It's also possible that MAP quantities varied between different sub-samples used by each laboratory from the same original individual animal, despite mixing of the original sample prior to creating sub-samples to be sent to laboratories B-D.

It's also plausible that other factors may have contributed to the wide range of apparent sensitivities resulting in the poor levels of inter-laboratory agreement. Potential explanations for the wide variation in results between laboratories include differences in sample volume, pooling methodology, DNA extraction protocols and primer pairs employed, all of which may affect sensitivity, as well as interpretation of results.

Stipulated individual faecal sample volumes for assays A-D were 1, 2, 5 and 1g respectively. Interestingly, both the most and least sensitive assays (A and D respectively) required the smallest volumes of faeces to be submitted which indicates variation in performance is not attributable to this factor alone. However, the low volume of faeces required for assay D may contribute to overall poor performance.

Selection of an appropriate DNA extraction kit is important for maximising both final eluted DNA concentrations and purity (23). Efficient extraction and DNA isolation is particularly challenging with both a complex biological matrix such as faeces and an organism such as MAP. Faeces contain a number of PCR-inhibiting substances such as polysaccharides, heme, plant components, phytic acid, polyphenols, bile salts and Ca²⁺ ions which can lead to false-negative results by inhibiting DNA amplification in the PCR (24) thus compromising the PCR reaction. Additionally, MAP bacteria possess a thick waxy cell wall with a high lipid content that renders them inherently difficult to be lysed, resulting in low DNA recovery and subsequently reduced PCR sensitivity (23, 25). Direct comparisons between performance of the extraction kits used by laboratories A-D was not possible here although it's feasible that this affected the overall diagnostic sensitivity.

[revised manuscript text omitted]

Authors: Worsley L^{1*}, Davies PL¹

Author affiliations: ¹ Department of Livestock & One Health, Institute of Infection Veterinary & Ecological Sciences, University of Liverpool, Leahurst Campus, Cheshire CH64 7TE.

*corresponding author: L.Taylor11@liverpool.co.uk

Abstract

Johne's disease is an infectious enteric disease caused by *Mycobacterium avium* subspecies *paratuberculosis* (MAP) affecting ruminant species worldwide. In Project 1, an independent performance comparison ring trail was conducted between three different commercial MAP quantitative polymerase chain reaction (qPCR) assay services (B, C and D) currently marketed in Great Britain by three separate laboratories against each other and against a fourth assay (A) (1) not currently available commercially in GB. A total of 205 individual ovine and bovine samples from 5 farms were analysed to give 41 sets of pooled results (pool size 5) from each laboratory according to their own specific protocols. The number of positive pools for assays A-D were 18, 12, 11 and 1 (43.9, 29.2, 26.8 and 2.4%) respectively. Assessment of interrater reliability resulted in a Fleiss' kappa coefficient of 0.15 indicating very poor overall agreement between the 4 laboratories. At the farm level, laboratories A-D diagnosed 4, 3, 2 and 1 flocks respectively as MAP positive. In Project 2, 38 pooled ovine samples from 10 flocks were analysed to compare performance of laboratories A and B. The number of positive results for laboratories A and B were 24 (63.1%) and 17 (44.7%) respectively (Cohen's kappa 0.54), indicating that laboratory A was more sensitive than B in line with results from Project 1. Variation between laboratories offering MAP qPCR assays is a significant concern and further work is warranted to validate and standardise the performance of assays between laboratories for both ovine and bovine samples.

Importance

Our study reports the findings of an inter-laboratory ring trail comparing the performance of four different qPCR assay services for the detection of *Mycobacterium avium* subspecies *paratuberculosis* (MAP) infection in cattle and sheep. MAP is the causative agent of Johne's disease (also known as paratuberculosis), a significant production-limiting disease in livestock populations with a worldwide distribution. The content of this paper is significant and novel as it is the first to highlight the marked variation between the diagnostic sensitivity and reproducibility of the three principal commercial laboratories offering MAP qPCR diagnostic and screening services in Great Britain. The low sensitivity and high degree of variability between the laboratories is of great concern and relevance to veterinary practitioners and livestock producers.

INTRODUCTION

Johne's disease (JD), or paratuberculosis, is an important production-limiting chronic wasting disease caused by *Mycobacterium avium* subspecies *paratuberculosis* (MAP) affecting predominantly ruminants with a worldwide distribution. The disease is chronic and insidious in nature with a long incubation period during which infected animals shed MAP in their faeces, intermittently and at low

levels initially (2, 3) ~~progressing~~ to persistent and heavy shedding as the ~~diseases~~ progresses towards
the clinical stages for those animals that are not capable of clearing the infection (4, 5). In addition to
negatively impacting animal welfare, JD is responsible for economic losses through reduced
productivity, increased replacement costs associated with premature culling and increased mortality
(6, 7). To this end, disease control programmes have been developed in many ~~countries~~ including
Great Britain (8). These programmes require robust and reliable diagnostic and screening tests to
establish herd or flock MAP infection status, to identify infected individual animals for ~~culling~~ and
make informed decisions on retention of their progeny, identify uninfected herds/ flocks that are free
from disease as a source of replacement stock and also to satisfy specific health scheme requirements.
Paratuberculosis also raises public health concerns with ongoing contention around MAP's suspected
role in ~~the development of Crohn's disease in humans~~. This ~~places additional emphasis on~~ the
importance of implementing on-farm JD control programmes (9).

Antemortem diagnosis of MAP infection in the early sub-clinical stages of ~~disease~~ is challenging due
to the ~~lack of~~ sufficiently sensitive and cost-effective tests. Additionally, ~~whilst~~ commercially
available MAP qPCR assays are validated for bovine samples in GB, the same laboratories do not
offer validated equivalent tests for ovine samples. The ~~general~~ consensus is that ~~faecal~~ qPCR and
culture are more sensitive than antibody ~~tests~~ especially in the early stages of ~~disease as MAP faecal~~
shedding precedes a detectable antibody response in sheep and cattle (10-12).

Antibody ELISA tests are most appropriate in the more advanced stages of disease associated with
increasingly discernible clinical signs (2). However, despite their low sensitivity in the early stages of
infection, ELISA assays are frequently employed as herd and flock screening tests because of their
practicality and low cost. Indeed, whole herd serological screening using ELISA tests ~~are~~ routinely
used for official JD control programmes for GB beef herds and also for sheep ~~flocks~~ albeit much less
commonly.

~~Whilst faecal~~ culture is still often considered 'gold standard' ~~largely~~ due to its very high specificity
(13) sensitivity estimates vary considerably ~~dependant~~ on the MAP strain in question and stage of
infection ~~and thus magnitude of faecal~~ shedding in the individual or population being tested (14, 15).
In contrast to C-strains which are relatively amenable to laboratory culture techniques, S-strains
display slow and fastidious growth ~~and are highly sensitive~~ to various decontamination procedures
and certain antibiotics (16, 17). As a result, it's plausible that the sensitivity of ~~faecal~~ culture is
underestimated where S strains predominate (15). How relevant this is to GB is unknown, given that
there is a ~~paucity~~ of information regarding circulating MAP strains in our sheep and cattle
populations. ~~Faecal~~ real-time qPCR assays have been shown to demonstrate superior sensitivity to
~~faecal~~ culture (15, 18-20). Plausible explanations for this include the ability of qPCR to detect both C-
and S- strains and ~~also to detect live as well as~~ dead MAP organisms (15).

To the author's knowledge, there have been no published independent trials comparing the
performance of the different MAP qPCR assays commercially available in GB. ~~The aim of this study~~
~~was~~ to establish the comparable performance of commercially available pooled ~~faecal~~-MAP qPCR
services ~~that are available~~ as screening tests to veterinarians in clinical practice. The inter-laboratory
ring trial study design encompasses the combined effect of intrinsic biochemical and extrinsic
procedural factors that influence the sensitivity and specificity of the testing service provided to the
clinician.

MATERIALS AND METHODS

Two projects were conducted as part of this inter-laboratory ring trial study. Project 1 compared the
performance of 3 pooled ~~faecal~~-MAP qPCR tests (assay services B, ~~C and D~~) as provided by the 3
prominent laboratories (B, ~~C and D~~ respectively) offering this assay in GB against each other and also

against a fourth PCR (A) (1) not currently available commercially in GB and performed at University
of Liverpool (laboratory A). Laboratories B and C are government ~~subsidised whilst laboratory D~~ is a
private commercial enterprise. Project 2, compared the performance of laboratory A against
laboratory B using ~~a further~~ set of samples.

**Sample Selection**

In Project 1, ~~faecal~~ samples originated from a convenience sample of 5 farms participating in a
concurrent MAP prevalence study (in press). Samples were initially selected based on sufficient mass
to allow division of the sample to provide adequate mass for each of the laboratories according to
their specific requirements. Random allocation ~~thereafter~~ gave a total of 205 individual samples,
grouped in 41 pools of 5 for analysis by each laboratory. A pool size of 5 was chosen as this is the
maximum pool size laboratories B, ~~C and D~~ offer as part of their commercial services. All 5 farms
were sheep and beef mixed species farms. The sample set comprised 50 bovine and 155 ovine
samples. The number of ovine and bovine pools tested from each farm ~~are~~ shown in Table 1.

*Table 1. Summary of total number of ovine and bovine pools analysed from farms V-Z.*

Farm ID	Number of Ovine Pools Tested	Number of Bovine Pools Tested	Total Number of Pools Tested
V	5	2	7
W	5	0	5
X	8	0	8
Y	7	4	11
Z	6	4	10
Total	31	10	41

Samples were collected between August and October 2021 by LW. All samples were kept at 4°C and
sent to the University of Liverpool within 2-3 days of being ~~collected~~ where they were stored at -
70°C. Individual samples were identified with a unique ~~id and~~ instructions on the individual pool
composition were provided to each laboratory to ensure the same pre-defined pools were tested by
each laboratory. Samples were shipped directly to laboratories B, ~~C and D from University of~~
Liverpool.

For Project 2, samples originated from a second, independent convenience sample of 190 ovine ~~faecal~~
samples collected from 10 flocks. These were pooled according to the same parameters as previously
described into 38 pools of 5 individuals per pool and were analysed by ~~laboratory A~~ and B only due to
limited sample mass. Both projects were performed as blind studies.

**Laboratory Assays**

**Laboratory A**

Samples were analysed by LW at the University of Liverpool research laboratories using the DNA
extraction kit and RL-PCR assay developed by Kawaji et al (1).

**Sample Preparation.** Individual samples ~~pooled~~ into groups of 5 in the laboratory by creating
individual ~~faecal suspensions prior to pooling as described by Mita et al (21).~~ Briefly, for each

~~individual~~ sample, 1g of ~~faeces~~ placed into a 50ml centrifuge tube containing 20ml of sterile distilled
water, mixed and shaken vigorously with a vortex for 30 sec before allowing the suspension to stand
for 30 min. Repeated for all samples contributing to a pool of 5. One ml of each ~~faecal~~ suspension
subsequently collected and pooled into a fresh 15ml ~~tube~~ followed by centrifugation at 900 g for 30
129 min at room temperature. All but approximately 1.5 ml of supernatant ~~removed~~, ~~faecal~~ pellet re-
130 suspended in ~~remaining supernatant and entire volume transferred to tube~~ containing zirconia beads
provided with Johne-PureSpin kit (FASMAC Ltd).

**DNA Extraction.** DNA extraction ~~performed~~ using the Johne-PureSpin kit (FASMAC Ltd);
according to the manufacturer's instructions. Briefly, 1ml of supernatant from prepared ~~faecal~~
suspension ~~transferred to bead~~ tube provided and centrifuged at 17k g for 6 min at room temperature.
After removal of the supernatant, 400µl of lysis buffer 1-A ~~added~~ and samples ~~pulverised~~ for 20 min
at 30Hz using a tissue lyser (Qiagen Ltd). Following centrifugation at 17K g for 6 min, ~~supernatant~~
~~transferred to 1.5ml tube~~ containing 200µl of lysis buffer 1-B and 75µl of extraction buffer 2. After
centrifugation at 17K g for 12 min, 500µl of supernatant ~~mixed~~ with 400µl of binding buffer 3.
~~Mixture transferred to spin~~ column and centrifuged at 13Kg for 1 min. Column washed once with
600µl of washing buffer 4, centrifuged at 13K g for 1 min, then placed into new 1.5ml tube. ~~Wash~~
~~step repeated if sample still discoloured after initial wash.~~ DNA samples ~~eluted~~ with 50µl of elution
buffer 5 by centrifugation at 13K g for 1 min and stored at -20°C if PCR could not be performed
~~same~~ day.

**PCR assay.** ~~Performed~~ using Reso-Light (RL) IS900 qPCR assay as ~~described by~~ Kawaji et al (1):
Briefly, reaction mixture comprised 5µl of template DNA, 25µl of 2x GeneAce RL qPCR mix
(Nippon Gene), 1µl of 25pmol of forward (IS900-3) and reverse (IS900-32) primers, 1µl of 0.4fg/µl
internal control (IC), and 0.5µl of 1U/µl uracil-DNA glycosylase (UDG; Nippon Gene) and 16.5µl of
nuclease-free water to make up 50µl total reaction volume. Reaction mixtures aliquoted with
QIAgility instrument (Qiagen Ltd) and qPCR runs performed on Rotor-Gene Q machine (Qiagen Ltd)
on SYBR reading setting using the following reaction program: initial incubation for UDG at 50°C
for 2 mins, followed by activation step of 10 mins at 95°C, then 45 cycles of PCR amplification at
95°C for 30 secs and 68°C for 60 secs. After PCR amplification, dissociation curve data ~~collected~~ for
analysing melting temperature (T_m) peaks. T_m peaks for the target (91.5 ± 1.5°C) indicated ~~presence~~
of MAP IS900, ~~whilst~~ T_m peak for the IC (85.5 ± 1.5°C) indicated a reaction without inhibition. If
neither T_m peak ~~detected~~, ~~result interpreted~~ as invalid due to PCR inhibition. DNA extracts ran in
singular with PCR positive and negative controls and DNA extraction negative controls.

**Laboratory B**

Samples were analysed according to laboratory B's specific protocols. ~~Detailed~~ specification of the
protocol is not in the public domain. ~~Based on personal communication with a representative of the~~
~~laboratory, the author's understanding of the protocol is as follows: 2g of faecal sample required from~~
~~each individual animal contributing to the pool of 5. Faecal material is pooled and 2g of composite~~
~~faecal~~ sample undergoes purification steps with water before centrifugation and resuspension in TE
buffer in preparation for magnetic bead DNA extraction using MagMax pathogen RNA/DNA kit
(Applied Biosystems). PCR ~~performed~~ in duplicate for 45 cycles using the primer pair MP10-1 and
~~MP11-11 that were~~ designed to target the MAP IS900 element³. Interpretation based on Ct values and
melt curve value (T_m) are as follows: Positive samples have a Ct value combined with a T_m of
between 87-90°C inclusive. Negative samples ~~either~~ have a Ct value >45 or a ~~Ct value~~ <45 combined
with an incorrect T_m product. ~~For all positive samples showing a Ct value of 35 or higher both~~
~~replicates need to be analysed together.~~ If the replicate is also ~~positive~~ it will be reported as such. If
the replicate is negative, then it should be reported as negative. If there is still a ~~discrepancy e.g.~~ 1
replicate has the correct T_m product and Ct value below ~~35~~ and the other replicate is clear negative,

then it is repeated a third ~~time~~ and that result is reported. Repeats are done on the extracted DNA from
the original extraction process. Samples are always reported as either positive or negative.

**Laboratory C**

Samples were analysed according to ~~laboratory C's~~ specific protocols. ~~Detailed~~ specification of the
protocol is not in the public domain. Based on personal communication with a representative of the
laboratory, the author's understanding of the protocol is as follows: 5g of ~~faecal~~ sample from each
~~individual~~ animal contributing to the pool of 5 is added to an EZ PREP bottle containing 30 ml of a
proprietary buffer (IDVet), then vortexed for 30 seconds. 2 ml from each ~~individual faecal~~ suspension
is then squeezed through a filter, located in the cap of the EZ Prep lid, into a container and the
resulting 10ml mixed by vortexing. 1.5 ml of the pooled suspension is added to the beads tube for
magnetic bead DNA extraction ~~performed~~ using the ID Gene Mag Fast extraction kit (Innovative
Diagnostics). PCR ~~performed~~ using the ID Gene Paratuberculosis Duplex kit (Innovative Diagnostics)
~~which~~ includes an internal control. DNA extracts are run in singular for 40 cycles. Interpretation ~~is as~~
~~follows~~: Ct < 33, record sample as positive, not re-tested. Ct value >40, record as ~~negative~~. Ct ≥33,
repeat from the beginning of the extraction process. If Ct ≥33 on repeat, ~~record~~ sample as positive. If
Ct value >40 on repeat, record ~~sample~~ as inconclusive. ~~For comparison to the other laboratories,~~
~~statistical analyses were performed in parallel with inconclusive results coded as both positive (C1)~~
~~and negative (C2).~~

**Laboratory D**

Samples were analysed according to ~~laboratory D's~~ specific protocols. ~~Detailed~~ specification of the
protocol is not in the public domain. Based on personal communication with a representative of the
laboratory, the author's understanding of the protocol is as follows: 1g of ~~faeces~~ from each individual
animal contributing to the pool of 5 is mixed into 30ml of water, of which 1ml is used for DNA
extraction using MagMAX CORE Mechanical Lysis kit (Applied Biosystems). PCR ~~performed~~ using
the VetMAX MAP Screening kit (Applied Biosystems). DNA extracts are run in singular for 45
cycles. Interpretation ~~is as follows~~: Ct < 37, record sample as positive, not re-tested. Ct value >45,
record as negative. Ct ≥ 37, repeat from the beginning of the extraction process if sufficient amounts
of ~~original faecal sample remains, otherwise~~ with saved material from part way through the extraction
process. If Ct < 37 on repeat, record as positive. If Ct ≥ 37 on repeat, record as inconclusive or
positive depending on clinical history. If Ct value >45 on repeat, repeat extraction and record ~~best~~ of 3
results.

**Statistical analysis**

Interrater reliability was tested using Cohen's and Fleiss' kappa coefficients in Minitab®19 Statistical
Software (Minitab, LLC) to assess pairwise and overall agreement between all ~~4 laboratories~~
~~respectively~~ at the pool level.

**RESULTS**

**Project 1**

The ~~number of~~ positive pools for assays A, B, C1, ~~C2~~ and D were 18, 12, 11, ~~2~~ and 1 (43.9, 29.3,
26.8, ~~4.9~~ and 2.4%) ~~respectively~~. Table 2 shows a heat map of the 41 raw results for each of the
laboratories for farms V-Z for both ovine and bovine samples.

Table 2 Heat map of the raw pooled results for laboratories A, B, C1, C2 and D for farms V-Z for both ovine
 (O) and B (bovine) samples (n=41). 1 = positive, 0 = negative, i = inconclusive.

Farm ID	Species	A	B	C1	C2	D
V	O	0	0	0	0	0
V	O	1	0	0	0	0
V	O	0	0	0	0	0
V	O	0	0	0	0	0
V	O	0	0	0	0	0
V	B	0	1	0	0	0
V	B	0	0	i	0	0
W	O	0	0	0	0	0
W	O	0	0	0	0	0
W	O	0	0	0	0	0
W	O	0	0	0	0	0
W	O	0	0	0	0	0
W	O	0	0	0	0	0
X	O	1	0	i	0	0
X	O	1	0	i	0	0
X	O	1	1	0	0	0
X	O	1	1	0	0	0
X	O	1	1	0	0	0
X	O	1	1	1	1	0
X	O	1	1	i	0	0
X	O	1	1	i	0	0
Y	O	0	0	0	0	0
Y	O	0	0	0	0	0
Y	B	0	0	i	0	0
Y	B	0	0	i	0	0
Y	O	0	0	0	0	0
Y	O	0	0	0	0	0
Y	O	1	0	0	0	0
Y	O	0	0	0	0	0
Y	O	1	1	0	0	0
Y	B	0	0	i	0	0
Y	B	0	0	0	0	0
Z	O	1	0	i	0	0
Z	O	1	1	0	0	0
Z	O	1	0	1	1	1
Z	O	1	0	0	0	0
Z	B	0	0	0	0	0
Z	B	0	0	0	0	0
Z	B	0	0	0	0	0
Z	B	1	1	0	0	0
Z	O	1	1	0	0	0
Z	O	1	1	0	0	0

Tables 3 shows the results of each laboratory for ovine and bovine pools.

Table 3 Summary of number of positive ovine (black) and bovine (red) pools as detected by laboratories A, B,
 C1, C2 and D for farms V-Z.

Farm ID	Total Number of Pools Tested	Laboratory				
		A	B	C1	C2	D
V	5, 2	1, 0	0, 1	0, 1	0, 0	0, 0
W	5, 0	0, 0	0, 0	0, 0	0, 0	0, 0
X	8, 0	8, 0	6, 0	5, 0	1, 0	0, 0
Y	7, 4	2, 0	1, 0	0, 3	0, 0	0, 0
Z	6, 4	6, 1	3, 1	2, 0	1, 0	1, 0
Grand Total	31, 10	17, 1	10, 2	7, 4	2, 0	1, 0

The Fleiss' kappa coefficient between all four laboratories for both bovine and ovine pools was 0.15
 ($p = 0.011$ and 0.008 for C1 and C2 respectively). Cohen's kappa scores ranged between -0.05 and
 0.66 when considering all 41 samples as shown in Table 4 .

Table 4 Agreement analysis between laboratories A, B C1, C2 and D for all ovine and bovine samples ($n=41$).

Laboratory	A	B	C1	C2	D
A		0.59 ¹ (0) ²	0.22 (0.062)	0.12(0.051)	0.06 (0.126)
B			-0.03(0.568)	0.07 (0.254)	-0.05 (0.743)
C1					0.13(0.047)
C2					0.66 (0)
D					

- 1. Cohen's kappa statistic
 2. p ($vs < 0$) of κ

The Fleiss' kappa coefficient between all four laboratories for ovine samples was 0.19 and 0.12 ($p =$
 0.005 and 0.0501) for C1 and C2 respectively. Cohen's kappa scores ranged between -0.06 and 0.65
 when considering all 31 samples, as shown in Table 5.

Table 5 Agreement analysis between laboratories A, B C1, C2 and D for ovine samples ($n=31$).

Laboratory	A	B	C1	C2	D
A		0.56 ¹ (0) ²	0.39 (0.003)	0.11(0.092)	0.05(0.178)
B			0.12 (0.248)	0.07 (0.290)	-0.06 (0.759)
C1					0.21(0.030)

C2					0.65 (0)
D					

- 1. Cohen's kappa statistic
 2. p ($vs < 0$) of κ

The Fleiss' kappa coefficient between all four laboratories for bovine samples was -0.10 ($p = 0.773$)
 and 0.16 ($p = 0.109$) for C1 and ~~C2~~ respectively. Cohen's kappa scores ranged between -0.36 and 0.62
 when considering all 10 samples, as shown in Table 6.

Table 6 Agreement analysis between laboratories A, B C1, C2 and D for bovine samples ($n=10$).

Laboratory	A	B	C1	C2	D
A		0.62 ¹ (0.018) ²	-0.19 (0.805)	0 (*)	0(*)
B			-0.36 (0.902)	0 (*)	0 (*)
C1					0(*)
C2					N/A
D					

- 1. Cohen's kappa statistic
 2. p ($vs < 0$) of κ

For diagnosis of MAP infection at the farm level, laboratory A detected 4 out of the 5 flocks as MAP
 positive, laboratory B detected 3, laboratory C detected 2 (for both interpretations of inconclusive
 results) and laboratory D only 1 flock as MAP ~~positive~~ as shown in Table 7.

Table 7 Number of positive ovine pools detected by each of Laboratories A-D for farms V-Z. Highlighted cells
 show positive flock diagnoses.

Farm ID	Number of Pools	A	B	C1	C2	D
V	5	1	0	0	0	0
W	5	0	0	0	0	0
X	8	8	6	5	1	0
Y	7	2	1	0	0	0
Z	6	6	3	2	1	1
Total	31	17	10	7	2	1

For diagnosis of MAP infection ~~on~~ the three beef herds in the study, laboratory A detected 1 herd as
 MAP positive, laboratories B and C1 detected 2 (but not the same two) ~~whilst~~ C2 and D did not detect
 any herds as MAP ~~positive~~ as shown in Table 8.

Table 8 Number of positive bovine pools detected by each of Laboratories A-D for farms V-Z. Highlighted cells
 show positive herd diagnoses.

Farm ID	Number of Pools	A	B	C1	C2	D
V	2	0	1	1	0	0

Y	4	0	0	3	0	0
Z	4	1	1	0	0	0
Total	10	1	2	4	0	0

Project 2

The ~~number of~~ positive pools detected by laboratories A and B were 24 (63.1%) and 17 (44.7%)
 ~~respectively with~~ a Cohen's kappa score of 0.54. These results were similar to the ovine results of
 ~~Project 1~~ whereby agreement between laboratories A and B was moderate. Further, laboratory A's
 assay was again more sensitive than laboratory B's ~~assay~~. When considering samples from both
 ~~project 1~~ and 2 together, the Cohen's kappa score for all (n=79) and ~~ovine only~~ samples (n=69) was
 0.57 for and ~~0.54~~ respectively.

~~For~~ diagnosis of MAP infection at the farm level, laboratory A diagnosed all 10 flocks as MAP
 ~~positive whilst~~ laboratory B detected just 6 ~~flocks~~ as MAP positive.

*Table 9 Summary of number of positive pools as detected by laboratories A and B for farms A-J.*

Farm	Number of pools tested	Number of positive pools	
		Laboratory A	Laboratory B
A	7	7	6
B	3	2	1
C	3	3	3
D	5	4	5
E	1	1	1
F	3	1	0
G	4	2	1
H	2	2	0
I	4	1	0
J	6	1	0
Total	38	24	17

Discussion

Agreement between the four laboratories varied greatly and was generally remarkably poor for both
 ovine and bovine samples. Several kappa values below zero were recorded between laboratories B-D,
 ~~which represents~~ agreement worse than expected by chance ~~alone~~, or disagreement. Low negative
 scores (0 to -0.10) may generally be interpreted as "no agreement" but large negative kappa values,
 such as ~~-0.36~~ seen between laboratories B and C1 for bovine samples, ~~represents~~ substantial
 disagreement between these two operators. The highest interrater reliability was consistently seen
 between laboratories C2 and ~~D and this~~ was attributable to the high proportion of negative pooled
 results for both of these laboratories in both species. The most likely explanation for this agreement is
 ~~poor~~ sensitivity of laboratory D and to a slightly lesser ~~extent~~ C2. These two ~~sets of~~ results evoke a
 false sense of high interrater reliability and should be interpreted ~~with caution~~. Unfortunately, ~~there~~
 ~~was~~ no 'gold standard' diagnostic ~~test~~ available to ~~assess~~ 'correctness' of the results in this ring trial.
 However, the high specificity performance demonstrated previously for the primer pairs employed by
 the assays performed by laboratories A (1) and B (18) confers a significant degree of confidence that
 positive results from these two assays are valid. For ovine samples, laboratory A detected the highest
 number of positive ~~pools~~ followed by laboratory B. Agreement between these two sets of results was
 moderate (0.56 and 0.62 kappa values for ovine and bovine ~~pools~~ respectively for Project 1, and 0.54

for Project 2) and broadly in line with the 0.6 kappa score level considered to be the minimum
acceptable interrater agreement for clinical and diagnostic settings (22). ~~For bovine samples,~~
~~laboratory C1 detected the highest number of positive pools followed by laboratory B.~~ However, the
agreement between these two sets of results for bovine samples was ~~remarkably poor~~ with a
significant negative kappa value (-0.36). Further, there wasn't sufficient ~~faecal mass to be able~~
~~to assess repeatability by each laboratory~~ but this would have been another interesting test performance
characteristic to investigate.

The clinical consequences of poor qPCR performance are highlighted in ~~the marked difference in~~
~~detection rate of MAP infection at the flock and herd level by the different laboratories.~~ Results from
this ring trial indicate that the chance of diagnosing MAP infection in a flock is significantly affected
by which laboratory is selected by submitting veterinary clinicians ~~as well as~~ the number of pools
analysed. This was also true for diagnosing MAP infection in beef herds which is potentially even
more concerning as laboratories B, ~~C and D~~ all claim to have done significantly more internal
validation and proficiency testing for MAP PCR on bovine versus ovine ~~faeces~~. However, the
relatively small sample size of bovine pools used in this ring trial ~~make it difficult~~ to assess the
significance of this observation.

~~Some degree of disagreement between the four laboratories was to be expected especially for samples~~
~~that contained low amounts of MAP DNA.~~ MAP is known to 'clump' in ~~faeces (18) and,~~ for
stochastic reasons, this non-homogenous distribution could explain a lower detection rate for ~~any~~
samples with low MAP levels. All ~~of the~~ four laboratories use qualitative qPCR ~~assays and thus~~
~~quantifying~~ bacterial load of the samples in this study was not possible to explore this hypothesis.
~~Low MAP DNA levels would have been exasperated by the pooling process and resulting dilution~~
~~factor.~~ It's also possible that MAP quantities varied between different sub-samples used by each
laboratory from the same original individual animal, despite ~~mixing of the original sample prior to~~
~~creating sub-samples to be sent to laboratories B-D.~~

It's also plausible that other factors may have contributed to the wide range of apparent sensitivities
resulting in the poor levels of inter-laboratory agreement. Potential explanations for the wide variation
in results between laboratories include differences in sample volume, pooling methodology, DNA
extraction ~~protocols~~ and primer pairs employed, all of which may affect ~~sensitivity, as well as~~
interpretation of results.

~~Stipulated individual faecal sample volumes for assays A-D were 1, 2, 5 and 1g respectively.~~
Interestingly, both the most and least sensitive assays (A and ~~D~~) respectively) required the smallest
volumes of ~~faeces to be submitted~~ which indicates variation in performance is not attributable to this
factor alone. However, the low volume of ~~faeces~~ required for assay D may contribute to overall poor
performance.

~~Selection of an appropriate DNA extraction kit is important for maximising both~~ final eluted DNA
concentrations and purity (23). Efficient extraction and DNA isolation ~~is~~ particularly challenging with
~~both~~ a complex biological matrix such as ~~faeces~~ and an organism such as MAP. ~~Faeces contain a~~

[revised manuscript text omitted]

407 Barwell R, Moreira MAS, Slana I, Koehler H, Singh SV, Yoo HS, Chávez-Gris G, Goodridge
408 A, Ocepek M, Garrido J, Stevenson K, Collins M, Alonso B, Cirone K, Paolicchi F, Gavey L,
Rahman MT, de Marchin E, Van Praet W, Bauman C, Fecteau G, McKenna S, Salgado M,
Fernández-Silva J, Dziedzinska R, Echeverría G, Seppänen J, Thibault V, Fridriksdottir V,
Derakhshandeh A, Haghkhal M, Ruocco L, Kawaji S, et al. 2019. Control of
paratuberculosis: who, why and how. A review of 48 countries. *BMC Vet Res* 15:198.
- 9. Waddell LA, Rajić A, Stärk KD, McEwen SA. 2016. The potential Public Health Impact of
*Mycobacterium avium* ssp. paratuberculosis: Global Opinion Survey of Topic Specialists.
*Zoonoses Public Health* 63:212-22.
- 10. P C, K A, R W. Shedding of organisms and sub-clinical effects on production in pre-clinical
Merino sheep affected with ovine paratuberculosis, p 126-31. *In* EJB M, MT C (ed),
Madison: International Association for Paratuberculosis,
- 11. Kawaji S, Begg DJ, Plain KM, Whittington RJ. 2011. A longitudinal study to evaluate the
diagnostic potential of a direct faecal quantitative PCR test for Johne's disease in sheep.
*Veterinary Microbiology* 148:35-44.
- 12. Yamamoto T, Murai K, Hayama Y, Kobayashi S, Nagata R, Kawaji S, Osaki M, Sakakibara
SI, Tsutsui T. 2018. Evaluation of fecal shedding and antibody response in dairy cattle

[revised manuscript text omitted]

487

488

Response to Reviewers

Reviewer #2 comments:

Please consider including the qPCR analysis curves, as they would provide visual evidence of the positivity of the reaction.

Have included an image of the melt curve analysis and expected report for PCR controls for assay performed by laboratory A. This was the only assay performed by the authors and the equivalent images are not available for laboratories B-D as these were performed externally.

Reviewer #3 comments:

1. At the end of line 14, What's number(1) mean? is it a reference number? The abstract mustn't contain any citations. (1) referred to a citation, has now been removed.
2. In line 15, GB is short for what, usually in the abstract doesn't write an abbreviation without the full name. Changed to Great Britain.
3. LW.....shorts for what? Laura Worsley, have changed all references to LW to 'first author'.
4. et.al must be written in italic. Have changed all 'et al.'s' to italics.
5. The Author should follow a unique style in reference citations, a number of references without year. All references updated and correct for ASM journal referencing style
6. (the author's understanding of the protocol) In line 177, is the author create this protocol or retrieve it from other literature? can you illustrate this point? The details of this protocol are proprietary and are not in the public domain/ published in the literature. I explain where the details of the protocol have come from in the first half of the sentence 'Based on personal communication with a representative of the laboratory, the author's understanding of the protocol is as follows...'. This is also the case for materials and methods describing Laboratory B and D protocols as the same circumstances apply.
7. The writing language had multi grammar mistakes, especially in tenses. All corrected.
8. In materials & methods, the target(specific gene or a region in the genome) in qPCR isn't clear also housekeeping genes. Target and housekeeping genes are not known/ in the public domain or published literature for laboratories B-D as this information is proprietary as mentioned in a previous comment. Target genes for primers are mentioned for assay A (line 146).

Reviewer #4 comments:

Need major revision

Have implemented all suggestions except the following:

Line 15: reject suggestion of changing 'currently' to 'now'.

Line 17: Reject suggestion of deleting 'own' and 'number of'.

Line 19: Reject suggestion of changing 'very poor' to 'inferior' as latter suggests comparison to something whereas we're just describing overall agreement between the assays in this sentence.

Line 36: Reject suggestion to delete 'degree of'.

Line 40: Reject suggestion of changing 'important' to 'essential'. It doesn't make sense for a disease to be 'essential'.

Line 54: Reject suggested change of 'the development of' to 'developing'.

Line 57: Reject suggested change of 'lack of' to 'need'.

Line 60: Reject suggestion of splitting sentence.

Line 72: Reject suggestion of splitting sentence.

Line 125: Reject suggested deletion of 'individual'.

Line 144: Reject suggestion to delete 'described by'.

Line 175: Reject suggested change to capital L for laboratory. Inconsistent with the rest of the reviewer's mark up.

Line 178 and 179: Reject suggested deletions of 'individual'.

Line 185: Reject suggested change of 'negative' to 'unfavorable'.

Line 187-189: Reject suggested sentence re-structure.

Line 191: Reject suggested change to capital L for laboratory. Inconsistent with the rest of the reviewer's mark up.

Line 211: Reject suggestion of deleting 'number of'.

Line 253: Reject suggestion of deleting 'number of'.

Line 257: Reject suggestion for hyphenating 'ovine-only'.

Line 259: Reject suggested change of 'For' to 'To'.

Line 273: Reject suggested deletion of 'sets of'.

Line 284: Reject suggestion of changing 'remarkably poor' to 'inferior' as latter suggests comparison to something else.

Line 311: Reject suggestion to change assays to assay. Needs to be plural as referring to multiple tests.

Line 324: Reject suggestion to change laboratories from plural to singular. Needs to be plural as referring to multiples.

Line 326: As above.

Line 346: Reject suggestion to split into two separate sentences.

Line 350: Reject suggestion to split into two separate sentences. Reviewer's suggestion changes meaning intended by authors.

Line 354: Reject suggested change from 'extremely poor' to 'low' and also deletion of 'for purpose'.

Line 357: Reject suggested hyphenation change as inconsistent with rest of text.

Line 361: Reject suggestion to split into two separate sentences.

Line 374: Reject suggested change from 'inferior' to 'low' as this sentence incorporates a comparison statement.

Line 377: Reject suggested change from 'important' to 'acute'. This is a chronic disease not an acute one and this is not the point the author's intended to make.

October 12, 2023

Dr. Laura Worsley
University of Liverpool
Chester High Road
Neston CH64 7TE
United Kingdom

Re: Spectrum02210-23R1 (Inter-laboratory ring trial to compare four quantitative polymerase chain reaction assays employed for detection of *Mycobacterium avium* subspecies paratuberculosis)

Dear Dr. Laura Worsley:

Link Not Available

Sincerely,

Sadjia Bekal

Journals Department
Reviewer comments:

Reviewer #4 (Comments for the Author):

Minor corrections required

Staff Comments:

Preparing Revision Guidelines

Please return the manuscript within 60 days; if you cannot complete the modification within this time period, please contact me. If you do not wish to modify the manuscript and prefer to submit it to another journal, please notify me of your decision immediately so that the manuscript may be formally withdrawn from consideration by Microbiology Spectrum.

Inter-laboratory ring trial to compare four quantitative polymerase chain reaction assays employed for detection of *Mycobacterium avium* subspecies *paratuberculosis*

Authors: Worsley L^{1*}, Davies PL¹

Author affiliations: ¹ Department of Livestock & One Health, Institute of Infection Veterinary & Ecological Sciences, University of Liverpool, Leahurst Campus, Cheshire CH64 7TE.

*corresponding author: L.Taylor11@liverpool.co.uk

[revised manuscript text omitted]

fecal suspension was subsequently collected and pooled into a fresh 15ml tube, followed by centrifugation at 900 g for 30 min at room temperature. All but approximately 1.5 ml of supernatant was removed, the fecal pellet re-suspended in the remaining supernatant, and the entire volume was transferred to a tube containing zirconia beads provided with Johne-PureSpin kit (FASMALD).

DNA Extraction. DNA extraction was performed using the Johne-PureSpin kit (FASMALD) according to the manufacturer's instructions. Briefly, 1ml of supernatant from prepared fecal suspension was transferred to the bead tube provided and centrifuged at 17k g for 6 min at room temperature. After removal of the supernatant, 400µl of lysis buffer 1-A was added, and samples were pulverised for 20 min at 30Hz using a tissue lyser (Qiagen Ltd). Following centrifugation at 17K g for 6 min, the supernatant was transferred to a 1.5ml tube containing 200µl of lysis buffer 1-B and 75µl of extraction buffer 2. After centrifugation at 17K g for 12 min, 500µl of supernatant was mixed with 400µl of binding buffer 3. The mixture was transferred to a spin column and centrifuged at 13K g for 1 min. Column washed once with 600µl of washing buffer 4, centrifuged at 13K g for 1 min, then placed into new 1.5ml tube. The wash step was repeated if the sample was still discoloured after the initial wash. DNA samples were eluted with 50µl of elution buffer 5 by centrifugation at 13K g for 1 min and stored at -20°C if PCR could not be performed the same day.

PCR assay. This was performed using a Reso-Light (RL) IS900 qPCR assay as described by Kawaji *et al.* (20). Briefly, reaction mixture comprised 5µl of template DNA, 25µl of 2x GeneAce RL qPCR mix (Nippon Gene), 1µl of 25pmol of forward (IS900-3) and reverse (IS900-32) primers, 1µl of 0.4fg/µl internal control (IC), and 0.5µl of 1U/µl uracil-DNA glycosylase (UDG; Nippon Gene) and 16.5µl of nuclease-free water to make up 50µl total reaction volume. Reaction mixtures aliquoted with QIAgility instrument (Qiagen Ltd) and qPCR runs performed on Rotor-Gene Q machine (Qiagen Ltd) on SYBR reading setting using the following reaction program: initial incubation for UDG at 50°C for 2 mins, followed by activation step of 10 mins at 95°C, then 45 cycles of PCR amplification at 95°C for 30 secs and 68°C for 60 secs. After PCR amplification, dissociation curve data were collected for analysing to analyze melting temperature (Tm) peaks. Tm peaks for the target (91.5 ± 1.5°C) indicated the presence of MAP IS900, while Tm peak for the IC (85.5 ± 1.5°C) indicated a reaction without inhibition. If neither Tm peak was detected, the result was interpreted as invalid due to PCR inhibition. DNA extracts ran in singular with PCR positive and negative controls and DNA extraction negative controls.

Figure 2. Melt curve analysis for the Reso-Light (RL) IS900 qPCR assay used by laboratory A. Shows melting temperature peaks for the target IS900 and internal control and expected results for the strong and weak positive and negative controls.

Laboratory B

Samples were analysed according to laboratory B's specific protocols. The detailed specification of the protocol is not in the public domain. Based on personal communication with a laboratory representative, the author's understanding of the protocol is as follows: 2g of fecal sample is required from each individual animal, contributing to the pool of 5. Fecal material is pooled, and 2g of composite fecal sample undergoes purification steps with water before centrifugation and resuspension in TE buffer in preparation for magnetic bead DNA extraction using MagMax pathogen RNA/DNA kit (Applied Biosystems). PCR is performed in duplicate for 45 cycles using the primer pair MP10-1 and MP11-11, designed to target the MAP IS900 element. Interpretation based on Ct values and melt curve value (T_m) are as follows: Positive samples have a Ct value combined with a T_m of between 87-90°C inclusive. Negative samples have a Ct value >45 or <45 combined with an incorrect T_m product. Both replicates need to be analysed together for all positive samples showing a Ct value of 35 or higher. If the replicate is also positive, it will be reported as such. If the replicate is negative, then it should be reported as negative. If there is still a discrepancy, e.g., 1 replicate has the correct T_m product and Ct value below 35, and the other replicate is clear negative, then it is repeated a third time, and that result is reported. Repeats are done on the extracted DNA from the original extraction process. Samples are always reported as either positive or negative.

Laboratory C

Samples were analysed according to laboratory Laboratory C's specific protocols. The detailed specification of the protocol is not in the public domain. Based on personal communication with a representative of the laboratory, the author's understanding of the protocol is as follows: 5g of fecal sample from each individual animal contributing to the pool of 5 is added to an EZ PREP bottle containing 30 ml of a proprietary buffer (IDVet), then vortexed for 30 seconds. 2 ml from each individual fecal suspension is then squeezed through a filter, located in the cap of the EZ Prep lid, into a container and the resulting 10ml mixed by vortexing. 1.5 ml of the pooled suspension is added to the beads tube for magnetic bead DNA extraction using the ID Gene Mag Fast extraction kit (Innovative Diagnostics). PCR is performed using the ID Gene Paratuberculosis Duplex kit (Innovative Diagnostics), which includes an internal control. DNA extracts are run in singular for 40 cycles. Interpretation: Ct < 33, record sample as positive, not re-tested. Ct value >40, record as negative. Ct ≥33, repeat from the beginning of the extraction process. If Ct ≥33 on repeat, record the sample as positive. If Ct >40 on repeat, record the sample as inconclusive. For comparison Compared to the other laboratories, statistical analyses were performed in parallel with inconclusive results coded as both positive (C1) and negative (C2).

Laboratory D

Samples were analysed according to laboratory Laboratory D's specific protocols. The detailed specification of the protocol is not in the public domain. Based on personal communication with a representative of the laboratory, the author's understanding of the protocol is as follows: 1g of feces from each individual animal contributing to the pool of 5 is mixed into 30ml of water, of which 1ml is used for DNA extraction using MagMAX CORE Mechanical Lysis kit (Applied Biosystems). PCR is performed using the VetMAX MAP Screening kit (Applied Biosystems). DNA extracts are run in singular for 45 cycles. Interpretation: Ct < 37, record sample as positive, not re-tested. Ct value >45, record as negative. Ct ≥ 37, repeat from the beginning of the extraction process if sufficient amounts of original fecal samples remain; otherwise, with saved material from part way through the extraction process. If Ct < 37 on repeat, record as positive. If Ct ≥ 37 on repeat, record as inconclusive or positive depending on clinical history. If the Ct value >45 on repeat, repeat extraction and record the best of 3 results.

Statistical analysis

Interrater reliability was tested using Cohen’s and Fleiss’ kappa coefficients in Minitab®19 Statistical Software (Minitab, LLC) to assess pairwise and overall agreement between all **four laboratories, respectively**, at the pool level.

RESULTS

Project 1

The **number of** positive pools for assays A, B, C1, **C2, and D** were 18, 12, 11, **2, and 1** (43.9, 29.3, 26.8, **4.9, and 2.4%**), **respectively**. Table 2 shows a heat map of the 41 raw results for each of the laboratories for farms V-Z for both ovine and bovine samples.

*Table 3 Heat map of the raw pooled results for laboratories A, B, C1, **C2, and D** for farms V-Z for both ovine (O) and B (bovine) samples (n=41). 1 = positive, 0 = negative, i = inconclusive.*

Farm ID	Species	A	B	C1	C2	D
V	O	0	0	0	0	0
V	O	1	0	0	0	0
V	O	0	0	0	0	0
V	O	0	0	0	0	0
V	O	0	0	0	0	0
V	B	0	1	0	0	0
V	B	0	0	i	0	0
W	O	0	0	0	0	0
W	O	0	0	0	0	0
W	O	0	0	0	0	0
W	O	0	0	0	0	0
W	O	0	0	0	0	0
W	O	0	0	0	0	0
X	O	1	0	i	0	0
X	O	1	0	i	0	0
X	O	1	1	0	0	0
X	O	1	1	0	0	0
X	O	1	1	0	0	0
X	O	1	1	1	1	0
X	O	1	1	i	0	0
X	O	1	1	i	0	0
Y	O	0	0	0	0	0
Y	O	0	0	0	0	0
Y	B	0	0	i	0	0
Y	B	0	0	i	0	0
Y	O	0	0	0	0	0
Y	O	0	0	0	0	0
Y	O	1	0	0	0	0
Y	O	0	0	0	0	0

Y	O	1	1	0	0	0
Y	B	0	0	i	0	0
Y	B	0	0	0	0	0
Z	O	1	0	i	0	0
Z	O	1	1	0	0	0
Z	O	1	0	1	1	1
Z	O	1	0	0	0	0
Z	B	0	0	0	0	0
Z	B	0	0	0	0	0
Z	B	0	0	0	0	0
Z	B	1	1	0	0	0
Z	O	1	1	0	0	0
Z	O	1	1	0	0	0

Table 3 shows the results of each laboratory for ovine and bovine pools.

Table 3 Summary of number of positive ovine (black) and bovine (red) pools as detected by laboratories A, B, C1, C2, and D for farms V-Z.

Farm ID	Total Number of Pools Tested	Laboratory				
		A	B	C1	C2	D
V	5, 2	1, 0	0, 1	0, 1	0, 0	0, 0
W	5, 0	0, 0	0, 0	0, 0	0, 0	0, 0
X	8, 0	8, 0	6, 0	5, 0	1, 0	0, 0
Y	7, 4	2, 0	1, 0	0, 3	0, 0	0, 0
Z	6, 4	6, 1	3, 1	2, 0	1, 0	1, 0
Grand Total	31, 10	17, 1	10, 2	7, 4	2, 0	1, 0

The Fleiss' kappa coefficient between all four laboratories for both bovine and ovine pools was 0.15 ($p = 0.011$ and 0.008 for C1 and C2, respectively). Cohen's kappa scores ranged between -0.05 and 0.66 when considering all 41 samples, as shown in Table 4.

Table 4 Agreement analysis between laboratories A, B C1, C2, and D for all ovine and bovine samples ($n=41$).

Laboratory	A	B	C1	C2	D
A	0.59¹(0)²	0.59 ¹ (0) ²	0.22 (0.062)	0.12(0.051)	0.06 (0.126)
B	0.22 (0.062)	0.59¹(0)²	-0.03(0.568)	0.07 (0.254)	-0.05 (0.743)
C1	0.12(0.051)	-0.03(0.568)	0.22 (0.062)	0.12(0.051)	0.13(0.047)
C2	0.06 (0.126)	-0.05 (0.743)	0.07 (0.254)	0.12(0.051)	0.66 (0)

D					
---	--	--	--	--	--

1. Cohen's kappa statistic
2. p (vs < 0) of κ

The Fleiss' kappa coefficient between all four laboratories for ovine samples was 0.19 and 0.12 ($p = 0.005$ and 0.0501) for C1 and C2, respectively. Cohen's kappa scores ranged between -0.06 and 0.65 when considering all 31 samples, as shown in Table 5.

Table 5 Agreement analysis between laboratories A, B C1, C2, and D for ovine samples ($n=31$).

Laboratory	A	B	C1	C2	D
A		0.56 ¹ (0) ²	0.39 (0.003)	0.11(0.092)	0.05(0.178)
B			0.12 (0.248)	0.07 (0.290)	-0.06 (0.759)
C1					0.21(0.030)
C2					0.65 (0)
D					

1. Cohen's kappa statistic
2. p (vs < 0) of κ

The Fleiss' kappa coefficient between all four laboratories for bovine samples was -0.10 ($p = 0.773$) and 0.16 ($p = 0.109$) for C1 and C2, respectively. Cohen's kappa scores ranged between -0.36 and 0.62 when considering all 10 samples, as shown in Table 6.

Table 6 Agreement analysis between laboratories A, B C1, C2, and D for bovine samples ($n=10$).

Laboratory	A	B	C1	C2	D
A		0.62 ¹ (0.018) ²	-0.19 (0.805)	0 (*)	0(*)
B			-0.36 (0.902)	0 (*)	0 (*)
C1					0(*)
C2					N/A
D					

1. Cohen's kappa statistic
2. p (vs < 0) of κ

For diagnosis of MAP infection at the farm level, laboratory A detected 4 out of the 5 flocks as MAP positive, laboratory B detected 3, laboratory C detected 2 (for both interpretations of inconclusive results) and laboratory D only 1 flock as MAP positive, as shown in Table 7.

Table 7 Number of positive ovine pools detected by each of Laboratories A-D for farms V-Z. Highlighted cells show positive flock diagnoses.

Farm ID	Number of Pools	A	B	C1	C2	D
V	5	1	0	0	0	0
W	5	0	0	0	0	0
X	8	8	6	5	1	0
Y	7	2	1	0	0	0
Z	6	6	3	2	1	1
Total	31	17	10	7	2	1

For diagnosis of MAP infection **in** the three beef herds in the study, laboratory A detected 1 herd as MAP positive, laboratories B and C1 detected 2 (but not the same two), **while** C2 and D did not detect any herds as MAP **positive, as** shown in Table 8.

Table 8 Number of positive bovine pools detected by each of Laboratories A-D for farms V-Z. Highlighted cells show positive herd diagnoses.

Farm ID	Number of Pools	A	B	C1	C2	D
V	2	0	1	1	0	0
Y	4	0	0	3	0	0
Z	4	1	1	0	0	0
Total	10	1	2	4	0	0

Project 2

The **number of** positive pools detected by laboratories A and B were 24 (63.1%) and 17 (44.7%), **respectively, with** a Cohen's kappa score of 0.54. These results were similar to the ovine results of **Project 1**, whereby agreement between laboratories A and B was moderate. Further, laboratory A's assay was again more sensitive than **laboratory B's**. When considering samples from **both projects** 1 and 2 together, the Cohen's kappa score for all (n=79) and **ovine-only** samples (n=69) was 0.57 for and **0.54, respectively**.

For To diagnosis **of** MAP infection at the farm level, laboratory A diagnosed all 10 flocks as MAP positive, **while** laboratory B detected just **6** as MAP positive.

Table 9 Summary of number of positive pools as detected by laboratories A and B for farms A-J.

Farm	Number of pools tested	Number of positive pools	
		Laboratory A	Laboratory B
A	7	7	6
B	3	2	1
C	3	3	3
D	5	4	5
E	1	1	1
F	3	1	0
G	4	2	1
H	2	2	0
I	4	1	0
J	6	1	0
Total	38	24	17

Discussion

Agreement between the four laboratories varied greatly and was generally remarkably poor for both ovine and bovine samples. Several kappa values below zero were recorded between laboratories B-D, representing agreement worse than expected by chance alone or disagreement. Low negative scores (0 to -0.10) may generally be interpreted as “no agreement,” but agreement.” Still, large negative kappa values, such as -0.36, seen between laboratories B and C1 for bovine samples, represent substantial disagreement between these two operators. The highest interrater reliability was consistently seen between laboratories C2 and D, and this which was attributable to the high proportion of negative pooled results for both of these laboratories in both species. The most likely explanation for this agreement is the poor sensitivity of laboratory D and, to a slightly lesser extent, C2. These two sets of results evoke a false sense of high interrater reliability and should be interpreted cautiously. Unfortunately, no ‘gold standard’ diagnostic test was available to assess the ‘correctness’ of the results in this ring trial. However, the high specificity performance demonstrated previously for the primer pairs employed by the assays performed by laboratories A (20) and B (17) confers a significant degree of confidence that positive results from these two assays are valid. For ovine samples, laboratory A detected the highest number of positive pools, followed by laboratory B. Agreement between these two sets of results was moderate (0.56 and 0.62 kappa values for ovine and bovine pools, respectively for Project 1 and 0.54 for Project 2) and broadly in line with the 0.6 kappa score level considered to be the minimum acceptable interrater agreement for clinical and diagnostic settings (22). Laboratory C1 detected the highest number of positive pools for bovine samples, followed by laboratory B. However, the agreement between these two sets of results for bovine samples was remarkably poor, with a significant negative kappa value (-0.36). Further, there wasn’t sufficient fecal mass to assess repeatability by each laboratory, but this would have been another interesting test performance characteristic to investigate.

The clinical consequences of poor qPCR performance are highlighted by the four laboratories’ marked differences in the detection of MAP infection at the flock and herd levels. The four laboratories’ marked differences in detecting MAP infection at the flock and herd levels highlight 
[revised manuscript text omitted]

- Barwell R, Moreira MAS, Slana I, Koehler H, Singh SV, Yoo HS, Chávez-Gris G, Goodridge A, Ocepek M, Garrido J, Stevenson K, Collins M, Alonso B, Cirone K, Paolicchi F, Gavey L, Rahman MT, de Marchin E, Van Praet W, Bauman C, Fecteau G, McKenna S, Salgado M, Fernández-Silva J, Dziedzinska R, Echeverría G, Seppänen J, Thibault V, Fridriksdottir V, Derakhshandeh A, Haghkhal M, Ruocco L, Kawaji S, Momotani E, Heuer C, Norton S, Cadmus S, Agdestein A, Kampen A, Sztejn J, Frössling J, Schwan E, Caldow G, Strain S, Carter M, Wells S, Munyeme M, Wolf R, Gurung R, Verdugo C, Fourichon C, Yamamoto T, Thapaliya S, Di Labio E, Ekgat M, Gil A, Nuñez Alesandre A, Piaggio J, Suanes A, H. de Waard J. 2019. Control of paratuberculosis: who, why and how. A review of 48 countries. *BMC Vet Res* 15:198.
8. Waddell LA, Rajić A, Stärk KD, McEwen SA. 2016. The potential Public Health Impact of *Mycobacterium avium* ssp. paratuberculosis: Global Opinion Survey of Topic Specialists. *Zoonoses Public Health* 63:212-22.
 9. Chaitaweesub P, Abbott KA, Whittington R, Marshall DJ. 1999. Shedding of organisms and sub-clinical effects on production in pre-clinical Merino sheep affected with ovine paratuberculosis, p 126-31. In Manning EJB, Collins MT (ed), *Proceedings of the Sixth International Colloquium on Paratuberculosis*. Madison, Wis: International Association for Paratuberculosis.
 10. Kawaji S, Begg DJ, Plain KM, Whittington RJ. 2011. A longitudinal study to evaluate the diagnostic potential of a direct faecal quantitative PCR test for Johne's disease in sheep. *Vet Microbiol* 148:35-44.
 11. Yamamoto T, Murai K, Hayama Y, Kobayashi S, Nagata R, Kawaji S, Osaki M, Sakakibara SI, Tsutsui T. 2018. Evaluation of fecal shedding and antibody response in dairy cattle infected with paratuberculosis using national surveillance data in Japan. *Prev Vet Med* 149:38-46.
 12. Whitlock RH, Wells SJ, Sweeney RW, Van Tiem J. 2000. ELISA and fecal culture for paratuberculosis (Johne's disease): sensitivity and specificity of each method. *Vet Microbiol* 77:387-98.
 13. Whittington RJ, Sergeant ES. 2001. Progress towards understanding the spread, detection and control of *Mycobacterium avium* subsp paratuberculosis in animal populations. *Aust Vet J* 79:267-278.
 14. Bauman CA, Jones-Bitton A, Jansen J, Kelton D, Menzies P. 2016. Evaluation of fecal culture and fecal RT-PCR to detect *Mycobacterium avium* ssp. paratuberculosis fecal shedding in dairy goats and dairy sheep using latent class Bayesian modeling. *BMC Vet Res* 12:212.
 15. Reddacliff LA, Vadali A, Whittington RJ. 2003. The effect of decontamination protocols on the numbers of sheep strain *Mycobacterium avium* subsp. paratuberculosis isolated from tissues and faeces. *Vet Microbiol* 95:271-82.
 16. Stevenson K, Alvarez J, Bakker D, Biet F, de Juan L, Denham S, Dimareli Z, Dohmann K, Gerlach GF, Heron I, Kopecna M, May L, Pavlik I, Sharp JM, Thibault VC, Willemsen P, Zadoks RN, Greig A. 2009. Occurrence of *Mycobacterium avium* subspecies paratuberculosis across host species and European countries with evidence for transmission between wildlife and domestic ruminants. *BMC Microbiol* 9:212.
 17. Kawaji S, Taylor DL, Mori Y, Whittington RJ. 2007. Detection of *Mycobacterium avium* subsp paratuberculosis in ovine faeces by direct quantitative PCR has similar or greater sensitivity compared to radiometric culture. *Vet Microbiol* 125:36-48.
 18. Arsenaault J, Singh Sohal J, Leboeuf A, Hélie P, Fecteau G, Robinson Y, L'Homme Y. 2019. Validation of an in-house real-time PCR fecal assay and comparison with two commercial assays for the antemortem detection of *Mycobacterium avium* subsp. paratuberculosis infection in culled sheep. *J Vet Diagn Invest* 31:58-68.
 19. Plain KM, Marsh IB, Waldron AM, Galea F, Whittington AM, Saunders VF, Begg DJ, de Silva K, Purdie AC, Whittington RJ. 2014. High-throughput direct fecal PCR assay for detection of *Mycobacterium avium* subsp. paratuberculosis in sheep and cattle. *J Clin Microbiol* 52:745-57.

20. Kawaji S, Nagata R, Minegishi Y, Saruyama Y, Mita A, Kishizuka S, Saito M, Mori Y. 2020. A Novel Real-time PCR-based Screening Test with Pooled Faecal Samples for Bovine Johne's Disease. *J Clin Microbiol* 58:e01761-20
21. Mita A, Mori Y, Nakagawa T, Tasaki T, Utiyama K, Mori H. 2016. Comparison of fecal pooling methods and DNA extraction kits for the detection of *Mycobacterium avium* subspecies paratuberculosis. *Microbiologyopen* 5:134-42.
22. McHugh ML. 2012. Interrater reliability: the kappa statistic. *Biochem Med (Zagreb)* 22:276-82.
23. Leite FL, Stokes KD, Robbe-Austerman S, Stabel JR. 2013. Comparison of fecal DNA extraction kits for the detection of *Mycobacterium avium* subsp. paratuberculosis by polymerase chain reaction. *J Vet Diagn Invest* 25:27-34.
24. Christopher-Hennings J, Dammen MA, Weeks SR, Epperson WB, Singh SN, Steinlicht GL, Fang Y, Skaare JL, Larsen JL, Payeur JB, Nelson EA. 2003. Comparison of two DNA extractions and nested PCR, real-time PCR, a new commercial PCR assay, and bacterial culture for detection of *Mycobacterium avium* subsp. paratuberculosis in bovine feces. *J Vet Diagn Invest* 15:87-93.
25. Logar K, Kopinč R, Bandelj P, Starič J, Lapanje A, Ocepek M. 2012. Evaluation of combined high-efficiency DNA extraction and real-time PCR for detection of *Mycobacterium avium* subsp. paratuberculosis in subclinically infected dairy cattle: comparison with faecal culture, milk real-time PCR and milk ELISA. *BMC Vet Res* 8:49.
26. Green EP, Tizard ML, Moss MT, Thompson J, Winterbourne DJ, McFadden JJ, Hermon-Taylor J. 1989. Sequence and characteristics of IS900, an insertion element identified in a human Crohn's disease isolate of *Mycobacterium paratuberculosis*. *Nucleic Acids Res* 17:9063-73.
27. Cousins DV, Whittington R, Marsh I, Masters A, Evans RJ, Kluver P. 1999. *Mycobacteria* distinct from *Mycobacterium avium* subsp. paratuberculosis isolated from the faeces of ruminants possess IS900-like sequences detectable IS900 polymerase chain reaction: implications for diagnosis. *Mol Cell Probes* 13:431-442.
28. Stabel JR, Bannantine JP. 2005. Development of a nested PCR method targeting a unique multicopy element, ISMap02, for detection of *Mycobacterium avium* subsp. paratuberculosis in fecal samples. *J Clin Microbiol* 43:4744-50.
29. Australia AH. Market Assurance Schemes, SheepMAP Accessed 23.11.2022.
30. Ly A, Dhand NK, Sergeant ESG, Marsh I, Plain KM. 2019. Determining an optimal pool size for testing beef herds for Johne's disease in Australia. *PLoS One* 14:e0225524.

Response to Reviewers

Reviewer #4 comments:

Minor corrections required

Have implemented all suggestions except the following:

Line 17: As previously, reject suggestion of deleting ‘number of’. Otherwise the sentence does not make sense.

Line 19: As previously, reject suggestion of changing ‘very poor’ to ‘inferior’ as latter suggests comparison to something whereas we’re just describing overall agreement between the assays in this sentence.

Line 52: Reject suggested change of ‘the development of’ to ‘developing’. Have deleted mention of development and simplified sentence to ‘MAP’s suspected role in Crohn’s disease’.

Line 155: Reject suggestion to delete ‘as’ between interpreted and invalid. Sentence doesn’t make sense without ‘as’.

Line 175: As previously, reject suggested change to capital L for laboratory. Inconsistent with the rest of the reviewer’s mark up.

Line 187-189: Reject suggested change. The reviewer has misunderstood the intended meaning of this sentence. I have changed the wording to help clarify.

Line 191: Reject suggested change to capital L for laboratory. Inconsistent with the rest of the reviewer’s mark up.

Line 201: Reject suggested change of including ‘the’ ahead of Ct value. Inconsistent with the rest of the reviewer’s mark up.

Line 211: As previously, reject suggestion of deleting ‘number of’. Otherwise the sentence does not make sense.

Line 253: As previously, reject suggestion of deleting ‘number of’. Otherwise the sentence does not make sense.

Line 257: As previously, reject suggestion for hyphenating ‘ovine-only’. Inconsistent with the rest of the reviewer’s mark up.

Line 259: As previously, reject suggested change of ‘For’ to ‘To’. Reviewer’s suggested change doesn’t make grammatical sense.

Line 268: Reject reviewer’s suggested change as it isn’t consistent with the author’s intended meaning.

Line 277: Reject reviewer’s suggestion of deleting ‘a’. Otherwise the sentence does not make sense.

Line 288: Slightly amended reviewer’s suggested sentence structure changes to help clarify.

Line 292: Reject reviewer’s suggestion of deleting ‘which is’. Otherwise the sentence does not make sense.

Line 297: Reject reviewer's suggestion of deleting 'degree of' as this loses some of the meaning intended by the author.

Line 301: Reject reviewer's suggestion of changing 'not possible' to 'impossible'. The latter sounds too dramatic.

Re: Spectrum02210-23R2 (Inter-laboratory ring trial to compare four quantitative polymerase chain reaction assays employed for detection of *Mycobacterium avium* subspecies paratuberculosis)

Dear Dr. Laura Worsley:

Your manuscript has been accepted, and I am forwarding it to the ASM production staff for publication. Your paper will first be checked to make sure all elements meet the technical requirements. ASM staff will contact you if anything needs to be revised before copyediting and production can begin. Otherwise, you will be notified when your proofs are ready to be viewed.

Sincerely,
Sadjia Bekal
Editor
Microbiology Spectrum